# LIGHT FORCING: Accelerating Autoregressive Video Diffusion via Sparse Attention

Chengtao Lv [1 2]  Yumeng Shi [1]  Yushi Huang [3]  Ruihao Gong [4 5]  Shen Ren [6]  Wenya Wang [1]

## Abstract

Advanced autoregressive (AR) video generation models have improved visual fidelity and interactivity, but the quadratic complexity of attention remains a primary bottleneck for efficient deployment. While existing sparse attention solutions have shown promise on bidirectional models, we identify that applying these solutions to AR models leads to considerable performance degradation for two reasons: isolated consideration of chunk generation and insufficient utilization of past informative context. Motivated by these observations, we propose LIGHT FORCING, the *first* sparse attention solution tailored for AR video generation models. It incorporates a *Chunk-Aware Growth* mechanism to quantitatively estimate the contribution of each chunk, which determines their sparsity allocation. This progressive sparsity increase strategy enables the current chunk to inherit prior knowledge in earlier chunks during generation. Additionally, we introduce a *Hierarchical Sparse Attention* to capture informative historical and local context in a coarse-to-fine manner. Such two-level mask selection strategy (i.e., frame and block level) can adaptively handle diverse attention patterns. Extensive experiments demonstrate that our method outperforms existing sparse attention in quality (e.g., 84.5 on VBench) and efficiency (e.g., 1.2∼1.3× end-to-end speedup). Combined with other efficient solutions, LIGHT FORCING further achieves a 2.0∼3.0× end-to-end speedup across diverse GPUs (e.g., 27.4 FPS on RTX 5090 and 33.9 FPS on H100). Code is released via this link.

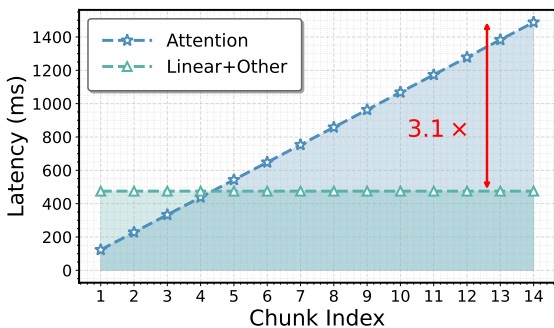

*Figure 1.* Runtime comparison of attention versus other components across chunk indices for Self Forcing (Huang et al., 2025a) 1.3B on RTX 5090. When the chunk index reaches 14, attention accounts for approximately ∼75% of the total latency.

## 1. Introduction

Recent notable advancements in video generation (Wan et al., 2025; Sun et al., 2024) have revolutionized artificial intelligence-generated content (AIGC). This progress can largely be attributed to the emergence of diffusion transformers (DiT) (Peebles & Xie, 2023), which leverage bidirectional attention to denoise all frames simultaneously. While video diffusion models (VDMs) can generate temporally consistent and long-duration videos, they struggle with temporal scalability, interactivity, and real-time deployment. In contrast, autoregressive (AR) video generation models naturally emerge as a more promising alternative, better suited to tackle these constraints. Moreover, recent AR models (Yin et al., 2025; Cui et al., 2025) replace lossy vector quantization techniques (Van Den Oord et al., 2017) with a chunk-by-chunk generation paradigm, yielding improved visual fidelity and interactivity. This also enables real-time applications in diverse downstream tasks, such as game simulation (Decart et al., 2024; Bruce et al., 2024; Parker-Holder et al., 2024) and robot learning (Yang et al., 2023; Li et al., 2025a).

Similar to bidirectional VDMs, the quadratic computational complexity of spatiotemporal 3D full attention in AR models still remains a major bottleneck for efficient deployment. As illustrated in Fig. 1, when generating a 480p video using Self-Forcing (Huang et al., 2025a) 1.3B, the attention

---
[1]Nanyang Technological University [2]AUMOVIO-NTU Corporate Lab [3]Hong Kong University of Science and Technology [4]Beihang University [5]Sensetime Research [6]AUMOVIO Singapore Pte Ltd. Correspondence to: Wenya Wang <wangwy@ntu.edu.sg>, Ruihao Gong <gongruihao@buaa.edu.cn>.

*Proceedings of the 43rd International Conference on Machine Learning*, Seoul, South Korea. PMLR 306, 2026. Copyright 2026 by the author(s).

consumes nearly three times the runtime of all other components combined (*i.e.*, linear layers, RoPE, *etc.*) at the last chunk. To mitigate the computational costs, one simple solution is to adopt various sparse attention methods introduced for bidirectional models (Zhang et al., 2025d; Xi et al., 2025; Yang et al., 2025b; Zhang et al., 2025b;a;c; Li et al., 2025c). These approaches mainly identify critical blocks utilizing either static (Zhang et al., 2025d; Li et al., 2025c) or dynamic (Wu et al., 2025a; Zhang et al., 2025c;a) sparse patterns in advance, and thus compute attention scores only for a small subset of tokens.

However, directly applying these sparse attention solutions to autoregressive (AR) models leads to significantly degraded generation quality compared to dense attention. We conduct in-depth investigations and observe that this performance drop arises from two primary aspects: ① sparse attention exacerbates the accumulation errors in AR models (*e.g.*, over-saturation in later chunks), while prior works largely ignore the heterogeneous contributions of different chunks to the global error accumulation. Our key insight is that, during denoising, the current chunk is essentially predicting the next noise level conditioned on past clean chunks. Therefore, later chunks are naturally prone to inheriting the quality of the past chunks. ② Another insight is the insufficient utilization of past key context. For each query block, the critical historical information varies significantly across model layers, attention heads, and denoising timesteps. However, existing methods (*e.g.*, sliding window attention (Beltagy et al., 2020) or adding chunk sinks (Yang et al., 2025a; Liu et al., 2025b)), inevitably discard part of this information, thereby harming long-range consistency and the richness of motion in generated videos.

Motivated by these findings, we propose LIGHT FORCING, an efficient variant specifically designed towards any autoregressive video generation models harnessing sparse attention. Specifically, ① we introduce a *Chunk-Aware Growth* (CAG) mechanism to quantitatively estimate the contributions of each chunk. Unlike chunk-agnostic policies that treat chunk generation in isolation, we view the generation of the current chunk as a further few-step denoising process conditioned on the previous clean chunk. From a theoretical perspective, we formulate the final sparsity allocation for each chunk as determined by its global accumulation error, which depends on two components (*i.e.*, the corresponding denoising steps and the score estimation). In other words, our method allocates lower sparsity priorities to earlier chunks, and progressively increases the sparsity in later chunks as they can inherit the structured knowledge stored in earlier chunks. ② We propose *Hierarchical Sparse Attention* (HSA), which preserves both global and local perception ability under a fixed computational budget. Specifically, HSA adopts a coarse-to-fine pipeline that selects sparse masks at both the frame and block levels for each query

block, enabling flexible and versatile attention modeling. This two-level strategy efficiently captures informative historical context while maintaining fast execution, thereby achieving an effective trade-off between model performance and computational cost.

We conduct extensive experiments to evaluate the effectiveness of our LIGHT FORCING. We compare our method with state-of-the-art sparse attention approaches on three autoregressive video generation models: Self Forcing (Huang et al., 2025a), LongLive (Yang et al., 2025a), and Infinite-Forcing (Junyi Chen, 2025). We report results on two benchmarks, VBench (Huang et al., 2024b) and VBench-Long (Huang et al., 2025c). The results show that our method consistently outperforms existing approaches in both generation quality and latency, and even surpasses dense attention in several metrics. For example, on Self Forcing (Huang et al., 2025a), our method achieves a total score of 84.5 while providing $1.3\times$ end-to-end and $3.79\times$ attention speedup. Furthermore, we provide complementary acceleration techniques, including FP8 linear layers, kernel fusion (*i.e.*, RoPE, RMSNorm, *etc.*), and an efficient VAE (*i.e.*, LightVAE (Contributors, 2025)). With plug-and-play configuration files, these optimizations enable approximately $2.0\sim3.0\times$ end-to-end speedup across diverse GPUs (*e.g.*, 27.4 FPS, 16.8 FPS, and 33.9 FPS on a single RTX 5090, A100, and H100 GPU, respectively).

To summarize, our main contributions are threefold:

- To the best of our knowledge, LIGHT FORCING is the first sparse attention solution specifically designed for autoregressive video generation models.
- We present *Chunk-Aware Growth (CAG)*. We allocate higher attention budgets to earlier chunks and progressively decay for later chunks, effectively reducing error propagation while preserving efficiency.
- We propose *Hierarchical Sparse Attention (HSA)*, which captures global and local dependencies via coarse-to-fine frame and block selection.
- Extensive experiments demonstrate the superior performance (*e.g.*, 84.5 on VBench) and real-time generation (*e.g.*, $2.0\sim3.0\times$ end-to-end speedup) of LIGHT FORCING across diverse GPUs.

## 2. Related Work

### 2.1. Autoregressive Video Diffusion

Compared with bidirectional video diffusion models (Wan et al., 2025; Yang et al., 2024; Sun et al., 2024) that denoise all frames jointly, autoregressive video generation models (Zhang et al., 2026; Huang et al., 2025a; Gu et al., 2025; Kodaira et al., 2025; Henschel et al., 2025) generate the next token or frame sequentially, and are thus inherently more suitable for real-time streaming applications.

Early approaches (Hu et al., 2024; Gao et al., 2024) adopted Teacher Forcing (TF), where training is conditioned on ground-truth tokens, but they suffer from reduced visual fidelity when generating long videos. Conversely, Diffusion Forcing (Chen et al., 2024) is trained with conditioning at arbitrary noise levels and has been adopted in models such as SkyReels-V2 (Chen et al., 2025a) and Magi-1 (Teng et al., 2025). CausVid (Yin et al., 2025) employs block causal attention, distilling a bidirectional teacher to a few-step causal student via distribution matching distillation (Yin et al., 2024). More recently, Self Forcing (Huang et al., 2025a) introduced a novel post-training paradigm that mitigates error accumulation arising from train-test misalignment. Subsequent works, including Rolling Forcing (Liu et al., 2025b), LongLive (Yang et al., 2025a), Self Forcing++ (Cui et al., 2025), and Reward Forcing (Lu et al., 2025), further address the limitation on the achievable generation length, object/scene dynamics or color drifts. Nevertheless, although autoregressive models with only a few denoising steps (*e.g.*, 4 steps) have substantially reduced latency, real-time generation on resource-constrained devices still remains challenging.

## 2.2. Sparse Attention

A large body of work (Wu et al., 2025a; Xu et al., 2025; Xi et al., 2025; Yang et al., 2025b; Zhang et al., 2025b; Wu et al., 2025b; Dalal et al., 2025) has explored how to alleviate the runtime bottleneck caused by quadratic-complexity attention in bidirectional video diffusion models, covering low-bit attention (Zhang et al., 2024b;a) and linear attention (Xie et al., 2024; Chen et al., 2025b; Huang et al., 2025b). Another promising line of work focuses on sparse attention, where approaches can be roughly categorized by whether they follow *static* or *dynamic* patterns to identify critical tokens with block-wise granularity. Static schemes (Zhang et al., 2025d; Li et al., 2025c; Hassani et al., 2025) usually prescribe sparsity masks via hand-crafted patterns, such as neighborhood (Hassani et al., 2025; Zhang et al., 2025d) or spatiotemporal structures (Xi et al., 2025). In contrast, dynamic solutions (Zhang et al., 2025b;a; Wu et al., 2025a; Cai et al., 2025) additionally introduce an online identification stage. These methods either utilize 1D (Zhang et al., 2025b; Wu et al., 2025a; Zhang et al., 2025a) or 3D (Zhang et al., 2025c; Wu et al., 2025a) mean pooling to aggregate blocks, and estimate their importance subsequently. Clustering-based strategies (Yang et al., 2025b) instead group semantically similar tokens together. In addition, several emerging *hybrid* attention mechanisms have been applied to video generation, including mixtures across different attention types (*e.g.*, combining linear attention and softmax attention (Zhang et al., 2025a)) and across different sparsity levels (*e.g.*, gating of twin-level (Zhang et al., 2025c) or pyramid-level sparse representation (Li et al., 2025b; Zhou et al., 2025)). However, the exploration of sparse attention for autoregressive video generation remains largely uncharted.

## 3. Preliminaries

**Autoregressive video diffusion modeling.** Autoregressive (AR) video diffusion models decompose video synthesis into *inter-chunk* autoregression and *intra-chunk* diffusion, combining the chain-rule factorization for temporal dependency modeling with the expressive denoising capability of diffusion models for high-fidelity frame generation. Specifically, given condition $c$, the joint distribution of an $N$-frame video sequence $\boldsymbol{x}^{1:N}$ is expressed as

$$p_\theta(\boldsymbol{x}^{1:N}|c) = \prod_{i=1}^{N} p_\theta(\boldsymbol{x}^i \mid \boldsymbol{x}^{<i}, c). \tag{1}$$

This formulation generates frames sequentially, where each conditional term $p_\theta(\boldsymbol{x}^i \mid \boldsymbol{x}^{<i}, c)$ is approximated by a few-step diffusion generator conditioned on KV cache (*i.e.*, previous clean frames) (Huang et al., 2025a; Cui et al., 2025; Yang et al., 2025a). Specifically, the conditional term $p_\theta(\boldsymbol{x}^i \mid \boldsymbol{x}^{<i}, c)$ can be defined as $f_{\theta,t_1} \circ f_{\theta,t_2} \circ \cdots \circ f_{\theta,t_T}(\boldsymbol{x}^i_{t_T})$, where $\boldsymbol{x}^i_{t_T} \sim \mathcal{N}(\boldsymbol{0}, \mathbf{I})$ and each transition is given by

$$f_{\theta,t_j}\left(\boldsymbol{x}^i_{t_j}\right) = \Psi\left(G_\theta(\boldsymbol{x}^i_{t_j}, t_j, \boldsymbol{x}^{<i}, c), \boldsymbol{\epsilon}_{t_{j-1}}, t_{j-1}\right), \tag{2}$$

where $G_\theta(\boldsymbol{x}^i_{t_j}, t_j, \boldsymbol{x}^{<i}, c)$ corresponds to the denoised estimate $\hat{\boldsymbol{x}}^i_0$, *i.e.*, a prediction of the clean chunk $i$ from the current noisy state $\boldsymbol{x}^i_{t_j}$ under the autoregressive context $\boldsymbol{x}^{<i}$ and condition $c$. The operator $\Psi(\cdot)$ denotes the forward corruption (re-noising) mapping that injects Gaussian noise at a lower noise level to produce the next state $\boldsymbol{x}^i_{t_{j-1}}$ for subsequent denoising. Advanced few-step AR video diffusion models often adopt the probability flow ODE formulation to define the forward noising trajectory and inject Gaussian noise (Song et al., 2023), *i.e.*, $(1 - \sigma_{t_{j-1}})\hat{\boldsymbol{x}}^i_0 + \sigma_{t_{j-1}}\boldsymbol{\epsilon}_{t_{j-1}}$, where $\boldsymbol{\epsilon}_{t_{j-1}} \sim \mathcal{N}(\boldsymbol{0}, \mathbf{I})$ and $\sigma_{t_{j-1}}$ controls the noise level.

**Blockwise sparse attention.** Practical sparse-attention systems enforce sparsity at the *block* (tile) granularity to better match modern accelerator hardware, enabling high utilization and efficient memory access patterns in GPU kernels such as FlashAttention (Dao, 2023). Concretely, given $\boldsymbol{q}, \boldsymbol{k}, \boldsymbol{v} \in \mathbb{R}^{n \times d}$, we partition[1] the sequence dimension into blocks and form

$$\boldsymbol{q} = \left[\boldsymbol{q}_1; \ldots; \boldsymbol{q}_{n_q}\right], \boldsymbol{k} = \left[\boldsymbol{k}_1; \ldots; \boldsymbol{k}_{n_k}\right], \boldsymbol{v} = \left[\boldsymbol{v}_1; \ldots; \boldsymbol{v}_{n_k}\right],$$

where $\boldsymbol{q}_i \in \mathbb{R}^{b_q \times d}$ and $\boldsymbol{k}_j, \boldsymbol{v}_j \in \mathbb{R}^{b_{kv} \times d}$, with $n_q = \lceil n/b_q \rceil$ and $n_k = \lceil n/b_{kv} \rceil$. We further define a *block mask* $\boldsymbol{B} \in$

---

[1] For simplicity, we assume $\boldsymbol{q}$ and $\boldsymbol{k}/\boldsymbol{v}$ have the same shape.

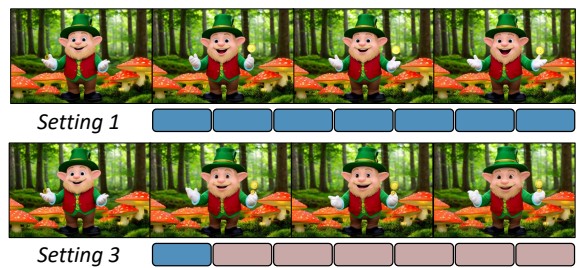 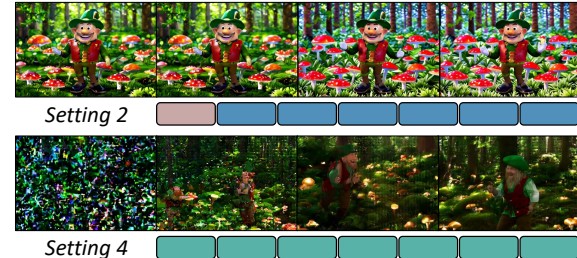

*Figure 2.* Comparison of different visual generation examples (*i.e.*, 7 chunks for 21 latent frames), where blue, red, and green boxes denote attention sparsity rates of 0%, 80%, and 90%, respectively.

$\{0, 1\}^{n_q \times n_k}$, where $\boldsymbol{B}_{ij} = 1$ indicates that the $(i, j)$ tile is active. Block-sparse attention can then be written as

$$\text{SparseAttn}(\boldsymbol{q}, \boldsymbol{k}, \boldsymbol{v}; \boldsymbol{B}) = \text{softmax}\left(\frac{\boldsymbol{q}\boldsymbol{k}^\top}{\sqrt{d_k}} \odot \boldsymbol{M}(\boldsymbol{B})\right)\boldsymbol{v}, \quad (3)$$

where $\boldsymbol{M}(\boldsymbol{B}) \in \{0, 1\}^{n \times n}$ expands $\boldsymbol{B}$ to an element-wise mask that is constant within each $(b_q \times b_{kv})$ tile, and $\odot$ denotes element-wise multiplication. Importantly, efficient implementations do not materialize $\boldsymbol{M}(\boldsymbol{B})$. Instead, they compute only the tile products $\boldsymbol{q}_i \boldsymbol{k}_j^\top$ and the corresponding value aggregation for indices $(i, j)$ with $\boldsymbol{B}_{ij} = 1$, skipping entire tiles when $\boldsymbol{B}_{ij} = 0$. Consequently, the computational and memory costs scale with the number of *active* blocks rather than $n^2$, while retaining GPU-friendly dense computation within each tile.

## 4. LIGHT FORCING

### 4.1. Chunk-Aware Growth Mechanism

Many acceleration techniques for bidirectional video diffusion models, including feature caching (Huang et al., 2024a; Liu et al., 2025a; Ma et al., 2024) and sparse attention (Li et al., 2025c; Zhang et al., 2025d), have observed pronounced sensitivity across different timesteps and layers. However, directly applying these *chunk-agnostic* policies of bidirectional models to few-step autoregressive video diffusion can be problematic: they ignore the heterogeneous contribution of different chunks to the *global accumulation error* that compounds over autoregressive rollout, and thus can easily trigger severe quality degradation or even collapse. To build intuition, we conduct several simple toy experiments that visually illustrate how generation behavior varies across chunks (as shown in Fig. 2).

First, we apply a moderately sparse attention ratio (*e.g.*, 80%) either to the first chunk (Setting 2) or to the subsequent chunks 2-7 (Setting 3). Surprisingly, we observe an interesting phenomenon: Setting 2 incurs an irreversible loss of visual quality (even over-saturation in the later chunks) that cannot be recovered even if later chunks revert to dense attention. Later frames in Setting 2 exhibit severe over-

saturation and exposure-bias artifacts. In contrast, Setting 3 achieves generation quality that is nearly lossless from Setting 1 even when only the first chunk is kept in dense attention. This further implies two observations: ① the first chunk acts as a *visual anchor* for the entire autoregressive rollout; ② once satisfactory priors are established in the first (or other early) chunk(s), subsequent chunks can readily inherit and propagate these priors with little difficulty. Therefore, we keep dense attention for the first chunk and allocate lower sparsity to other early chunks, while later chunks can tolerate higher sparsity. A natural question then arises: *Can we quantitatively allocate the sparsity budget across chunks?*

**Sparsity-Induced Error.** To solve the problem, we further explore generation performance under a radical sparsity rate (*i.e.*, 90%) in Setting 4, and observe that as time progresses, later chunks gradually become clearer compared to the initially Gaussian-noise-like appearance, suggesting that $G_\theta(\boldsymbol{x}_{t_j}^i, t_j, \boldsymbol{x}^{<i}, c)$ performs denoising toward the next noise level compared with $\boldsymbol{x}^{i-1}$. Moreover, we posit that $G_\theta(\boldsymbol{x}_{t_j}^i, t_j, \boldsymbol{x}^{<i}, c)$ essentially continues denoising for $T$ additional steps starting from the noisy level of $\boldsymbol{x}^{i-1}$ (verified in the Appendix).

These analyses reveal that sparsity effects can be reflected by the noise level in the final generated clean chunk $\boldsymbol{x}_{t_0}^i$ $(i = 1, \ldots, N)$. To capture this, we measure the sparsity-induced error of chunk $i$ as the variation distance $\text{TV}(\cdot, \cdot)$ between the clean data distribution $p$ and the generated distribution of $\boldsymbol{x}_{t_0}^i$, denoted $q_t$, with its noise level:

$$\text{TV}(q_t, p) \le C_1 \frac{d^2 \log^3 T}{\sqrt{T}} + C_2 \sqrt{d}\, \varepsilon_{\text{score}} \log^2 T, \quad (4)$$

where $C_1$, $C_2$, $\log^2 T$, $\log^3 T$ are constants[2] do not affect the asymptotic complexity. From this inequality, we can interpret the first term as the finite-step sampling error, which is $\propto 1/\sqrt{T}$, while the second term captures the effect of score estimation error, reflecting the approximation error

---

[2]Logarithmic factors of the form $\log^k T$ arise only as technical amplification constants in the proof and do not affect the asymptotic complexity.

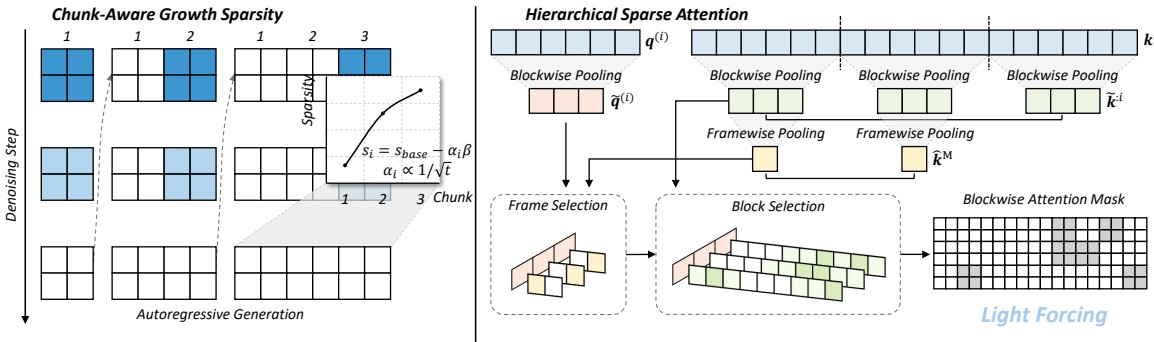

*Figure 3.* Overview of LIGHT FORCING. The left subfigure illustrates our *Chunk-Aware Growth* (Sec. 4.1) strategy for sparsity allocation across different chunks. The right subfigure demonstrates how *Hierarchical Sparse Attention* (Sec. 4.2) is utilized to efficiently retrieve long-range historical context. Note that a chunk corresponds to a group of frames processed in a single generation (*e.g.*, 3 frames in practice). For simplicity, we visualize each chunk as a single frame in the overview.

induced by imperfect model learning.

**Sparsity Allocation.** Intuitively, we should lower the sparsity ratio for chunks with more errors to preserve generation quality. Leveraging this insight, we propose a *Chunk-Aware Growth* (CAG) strategy that considers both the finite-step sampling error (Term 1) and the score estimation error (Term 2). In practice, we set dense attention for the first chunk to preserve the initial visual anchor, and apply CAG to the remaining chunks. For chunk $i > 1$, the sparsity ratio $s_i$ can be written as

$$s_i = s_{base} - \alpha_i \beta \quad (5)$$

where $\alpha_i$ denotes the noise level reached by the $i$-th chunk and scales as $\propto 1/\sqrt{T}$. The hyperparameter $s_{base}$ is a predefined constant that reflects the score estimation error. To solve for the modulated sparsity factor $\beta$, we enforce the total FLOPs after chunk-wise modulation to be equal to the FLOPs specified by the target sparsity ratio:

$$(1 - s_{target}) \sum_{i=2}^{n} l_i^q l_i^k d = \sum_{i=2}^{n} \left(1 - s_{base} + \alpha_i \beta\right) l_i^q l_i^k d, \quad (6)$$

where $s_{target}$ denotes the target sparsity ratio, and $l_i^q$ and $l_i^k$ denote the query and key sequence lengths for chunk $i$, respectively[3]. This equality yields $\beta$ and consequently determines the chunk-wise sparsity ratio $s_i$.

### 4.2. Hierarchical Sparse Attention

Another key challenge of autoregressive video generation models is that the number of historical frames grows linearly over time, making attention increasingly time-consuming and slowing down later-chunk generation. Many approaches (Liu et al., 2025b; Lu et al., 2025; Yang et al.,

2025a) mitigate this by adopting sliding-window attention (Beltagy et al., 2020) that truncates past frames to a fixed context length. While this alleviates the latency growth, it can induce history forgetting, leading to poor long-range consistency and repetitive motions in subsequent chunks. We believe that treating nearby frames as keyframes is suboptimal, since the historical frames that the current query is interested in vary across different layers, heads, and timesteps. As illustrated in Fig. 4, attention between historical frames exhibits complex patterns such as diagonal, attention-sink structures, which makes it difficult for a sliding window scheme to cover all informative context.

Inspired by these findings, we propose the *Hierarchical Sparse Attention* (HSA), which follows a coarse-to-fine paradigm for sparse attention on autoregressive video generation models. Specifically, each query block first retrieves a set of keyframes and then performs dynamic sparse attention over the selected frames. This dual-stage strategy not only bounds attention to a fixed computational complexity but also mitigates long-video consistency degradation and history-forgetting issues.

Formally, the process of generating chunk $i$ can be understood as computing attention between the query $q^{(i)}$ and the key-value pairs $\{k^{:i}, v^{:i}\}$. Here, $q^{(i)} \in \mathbb{R}^{(f \times n) \times d}$ and $\{k^{:i}, v^{:i}\} \in \mathbb{R}^{(i \times f \times n) \times d}$, where $f$ and $n$ denote the number of frames per chunk and the number of tokens per frame, respectively. To avoid computing attention between all key-value pairs, our HSA mainly consists of three components, *i.e.*, Token Compression, Mask Selection, and Blockwise Sparse Attention.

**Token Compression.** We first compress $q^{(i)}$ at the blockwise granularity and $k^{:i}$ at the blockwise and framewise

---

[3]For chunk $i$, given $q_i \in \mathbb{R}^{l_i^q \times d}$ and $k_i \in \mathbb{R}^{l_i^k \times d}$, the dominant FLOPs of attention are $\mathcal{O}(l_i^q l_i^k d)$.

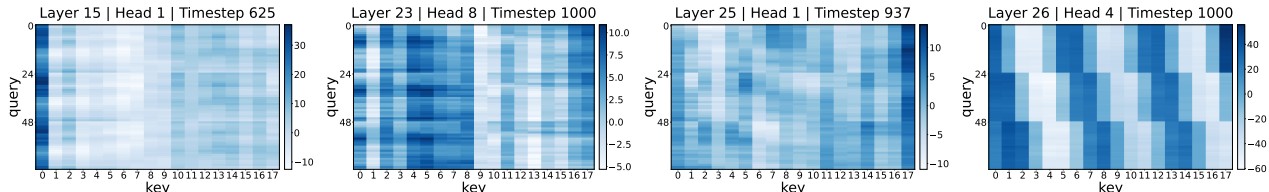

*Figure 4.* Visualization of attention logits between query blocks at chunk 7 (*i.e.*, frame 18-20, 24 blocks per frame) and all past key frames (*i.e.*, frame 0-17) on Self Forcing (Huang et al., 2025a).

granularity, which can be written as

$$\text{blockwise:} \quad \tilde{\boldsymbol{q}}^{(i)} = \phi\left(\boldsymbol{q}^{(i)}, b_q\right), \tilde{\boldsymbol{k}}^{:i} = \phi\left(\boldsymbol{k}^{:i}, b_{kv}\right),$$
$$\text{framewise:} \quad \hat{\boldsymbol{k}}^{\mathcal{M}} = \phi\left(\tilde{\boldsymbol{k}}^{\mathcal{M}}, \lceil n/b_{kv} \rceil\right), \tag{7}$$

where $\phi(\mathbf{x}, b)$ denotes a mean pooling operator that aggregates sequential tokens with size $b$. After applying this operation, the final compressed representations have shapes $\tilde{\boldsymbol{q}}^{(i)} \in \mathbb{R}^{(f \times \lceil n/b_q \rceil) \times d}$, $\tilde{\boldsymbol{k}}^{:i} \in \mathbb{R}^{(i \times f \times \lceil n/b_{kv} \rceil) \times d}$, and $\hat{\boldsymbol{k}}^{\mathcal{M}} \in \mathbb{R}^{m \times d}$. It is worth noting that $\hat{\boldsymbol{k}}^{\mathcal{M}}$ does not cover all past frames[4], since the sink frames, the nearest frames within the sliding window, and the key-value frames within *current* chunk $i$ are always selected.

**Mask Selection.** Based on the compressed representations, we perform a hierarchical mask selection in a coarse-to-fine manner. Specifically, for each query block in chunk $i$, we first retrieve a small set of relevant historical frames using the framewise-compressed keys, and then select critical blocks within the retrieved frames using blockwise-compressed keys.

Let $\tilde{\boldsymbol{q}}_r^{(i)} \in \mathbb{R}^d$ denote the $r$-th blockwise query summary in the current chunk, where $r \in \{1, \ldots, f \times \lceil n/b_q \rceil\}$. We compute frame-level relevance scores between the query block and each candidate past key in $\mathcal{M}$ using the framewise-compressed keys $\hat{\boldsymbol{k}}^{\mathcal{M}}$ as

$$p_r^{(i)} = \langle \tilde{\boldsymbol{q}}_r^{(i)}, \hat{\boldsymbol{k}}^{\mathcal{M}} \rangle \in \mathbb{R}^m, \tag{8}$$

where $p_r^{(i)}$ denotes the vector of frame-level logits, whose entries are given by the inner products between the query block summary $\tilde{\boldsymbol{q}}_r^{(i)}$ and each framewise-compressed key in $\hat{\boldsymbol{k}}^{\mathcal{M}}$. We then select the most relevant candidate past keys from $\mathcal{M}$:

$$\mathcal{T}_r = \text{TopK}_{\text{idx}}\left(p_r^{(i)}\right) \cup \mathcal{F}^{(i)}, \tag{9}$$

where $\text{TopK}_{\text{idx}}$ returns indices of the most relevant top-$k$ candidate frames and $\mathcal{F}^{(i)}$ denotes the always-selected

---
[4] $m = (i-1) \times f - n_{sink} - n_{win}$, where $n_{sink}$ denotes the number of earliest sink frames and $n_{win}$ denotes the small number of frames in the sliding window.

frame set discussed above, including a few sink frames, nearest frames, and the frames within the current chunk $i$, ensuring motion smoothness and full visibility over intra-chunk temporal dependencies.

Given the selected frame set $\mathcal{T}_r$, we further perform fine-grained blockwise selection among all blocks in the selected frames. For a frame $\tau \in \mathcal{T}_r$, let $\tilde{\boldsymbol{k}}_j^{(\tau)} \in \mathbb{R}^d$ denote the $j$-th blockwise key summary. We compute block-level relevance logits as

$$o_r^{(i)}(\tau, j) = \langle \tilde{\boldsymbol{q}}_r^{(i)}, \tilde{\boldsymbol{k}}_j^{(\tau)} \rangle, \tag{10}$$

and select the top-$k$ block pairs from the candidate set $\mathcal{B}_r$:

$$\mathcal{B}_r = \{(\tau, j) \mid \tau \in \mathcal{T}_r, \ j \in \{1, \ldots, \lceil n/b_{kv} \rceil\}\},$$
$$\mathcal{J}_r = \text{TopK}_{\text{idx}}\left(\{o_r^{(i)}(\tau, j)\}_{(\tau,j) \in \mathcal{B}_r}\right). \tag{11}$$

**Blockwise Sparse Attention.** Based on the selected frames and blocks, we construct a block-level attention mask $\boldsymbol{B}^{(i)} \in \{0, 1\}^{n_q \times n_{kv}}$. For the $r$-th query block, we have

$$\boldsymbol{B}_r^{(i)}(\tau, j) = \mathbb{1}\left[\tau \in \mathcal{T}_r, \ j \in \mathcal{J}_r(\tau)\right]. \tag{12}$$

Here, $n_q$ and $n_{kv}$ denote the number of query and key blocks, respectively. The final attention for the $r$-th query block is computed using blockwise sparse attention:

$$\text{Attn}_r^{(i)} = \text{softmax}\left(\frac{\boldsymbol{q}_r^{(i)}(\boldsymbol{k}^{:i})^\top}{\sqrt{d}} \odot \boldsymbol{M}(\boldsymbol{B}_r^{(i)})\right) \boldsymbol{v}^{:i}. \tag{13}$$

In summary, our HSA maintains a fixed attention complexity (independent of the total number of historical frames) while alleviating long-range consistency degradation. Meanwhile, our dual-stage mask selection incurs only a negligible overhead compared to conventional dynamic sparse attention, as it merely adds a frame-retrieval step (approximately a 2% increase in end-to-end runtime). CAG and HSA are complementary: CAG allocates a sparsity ratio for each chunk at a macro level, while HSA determines from a fine-grained perspective how much historical information each block in the current chunk can leverage.

*Table 1.* Performance comparison with state-of-the-art baselines on VBench (Huang et al., 2024b).

| Method | Latency (s)↓ | Speedup↑ | Aesthetic Quality↑ | Imaging Quality↑ | Motion Smoothness↑ | Dynamic Degree↑ | Subject Consistency↑ | Background Consistency↑ | Quality Score↑ | Semantic Score↑ | Total Score↑ |
|---|---|---|---|---|---|---|---|---|---|---|---|
| *Self-Forcing 1.3B* (`fps = 16`) | | | | | | | | | | | |
| FlashAttention2 (Dao, 2023) | 9.61 | 1.00× | 67.4 | 70.0 | 98.3 | 63.1 | 95.3 | 96.5 | 84.8 | 81.2 | 84.1 |
| STA (Zhang et al., 2025d) | 8.27 | 1.16× | 64.5 | 71.7 | 98.5 | 48.9 | 96.3 | 96.9 | 84.0 | 82.1 | 83.6 |
| Radial (Li et al., 2025c) | 7.39 | 1.30× | 45.8 | 66.1 | 96.0 | 88.6 | 90.2 | 93.6 | 78.7 | 53.7 | 73.7 |
| SVG2 (Yang et al., 2025b) | 21.38 | 0.45× | 66.0 | 68.2 | 97.8 | 72.8 | 93.6 | 95.6 | 83.9 | 78.5 | 82.8 |
| VMoBA (Wu et al., 2025a) | 7.42 | 1.29× | 65.2 | 69.9 | 97.3 | 84.2 | 92.8 | 95.5 | 84.5 | 80.3 | 83.6 |
| SLA (Zhang et al., 2025a) | 7.71 | 1.25× | 66.7 | 69.8 | 98.3 | 44.2 | 95.6 | 96.7 | 83.4 | 82.5 | 83.2 |
| LIGHT FORCING | **7.39** | **1.30×** | 67.2 | 71.0 | 98.3 | 66.7 | 96.2 | 96.5 | 85.4 | 80.9 | **84.5** |
| *LongLive 1.3B* (`fps = 16`) | | | | | | | | | | | |
| FlashAttention2 (Dao, 2023) | 10.47 | 1.00× | 68.7 | 69.3 | 98.8 | 39.2 | 97.0 | 97.2 | 83.8 | 80.7 | 83.2 |
| STA (Zhang et al., 2025d) | 9.56 | 1.10× | 65.6 | 71.2 | 99.0 | 22.8 | 97.4 | 97.8 | 82.8 | 81.6 | 82.6 |
| Radial (Li et al., 2025c) | 8.89 | 1.18× | 55.1 | 72.0 | 98.0 | 25.0 | 77.6 | 88.9 | 75.0 | 66.6 | 73.3 |
| SVG2 (Yang et al., 2025b) | 22.12 | 0.47× | 66.7 | 67.0 | 98.5 | 44.4 | 95.3 | 96.1 | 82.7 | 78.7 | 81.9 |
| VMoBA (Wu et al., 2025a) | 8.88 | 1.18× | 59.9 | 68.2 | 97.5 | 50.6 | 58.3 | 80.9 | 71.5 | 70.7 | 71.3 |
| LIGHT FORCING | **8.81** | **1.19×** | 67.2 | 70.6 | 98.2 | 59.4 | 96.9 | 96.7 | 84.8 | 80.2 | **83.9** |

## 5. Experiments

### 5.1. Experimental Details

**Implementation.** We build sparse attention on top of the currently open-sourced autoregressive video generation models, Self Forcing (Huang et al., 2025a), LongLive (Yang et al., 2025a) and Infinite-Forcing (Junyi Chen, 2025). Following them, we use a chunk size of three latent frames. For Finetunable methods (*i.e.*, VMoBA (Wu et al., 2025a), SLA (Zhang et al., 2025a) and LIGHT FORCING), we perform extra post-training for 2,000 iterations based on their pre-trained weights. For latency evaluation, we adopt the SpargeAttention kernel (Zhang et al., 2025b) for all methods (except Sparse VideoGen2 (Yang et al., 2025b) due to its variable block lengths, for which we use FlashInfer (Ye et al., 2025) as the inference backend), and report the measured latency on an RTX 5090 GPU.

**Evaluation.** We use VBench (Huang et al., 2024b) and VBench-Long (Huang et al., 2025c) to evaluate generation quality on 5/15-second videos across 16 dimensions. These dimensions include Subject Consistency, Background Consistency, Aesthetic Quality, Imaging Quality, Object Class, Multiple Objects, Color, Spatial Relationship, Scene, Temporal Style, Overall Consistency, Human Action, Temporal Flickering, Motion Smoothness, Dynamic Degree, and Appearance Style. We report a representative subset of these metrics in the main paper, and the complete results are reported in the appendix. Notably, we adopt the test prompts rewritten by Self Forcing (Huang et al., 2025a). For fair comparisons, we set the block size of all sparse attention methods to 64. We also adjust the resolution from $480 \times 832$ to $512 \times 768$, which avoids excessive padding overhead and potential non-equivalence introduced by certain methods

(*e.g.*, VMoBA (Wu et al., 2025a)).

**Baselines.** We compare LIGHT FORCING with state-of-the-art sparse attention methods for bidirectional video generation models, covering static mask selection approaches (STA (Zhang et al., 2025d), Sparse VideoGen2 (Yang et al., 2025b), and Radial Attention (Li et al., 2025c)) as well as dynamic mask selection approaches (VMoBA (Wu et al., 2025a) and SLA (Zhang et al., 2025a)). To ensure a fair comparison, we set the sparsity ratio of all static methods to around 80% (except for STA), and that of all dynamic methods to around 90%. Additional implementation details and hyperparameter settings for the specific method are provided in the Appendix.

### 5.2. Main Results

Tab. 1 reports the evaluation results of our LIGHT FORCING and state-of-the-art baselines on two mainstream autoregressive video generation models, *i.e.*, Self Forcing (Huang et al., 2025a) and LongLive (Yang et al., 2025a). We observe that LIGHT FORCING outperforms these methods by a large margin on most metrics (*e.g.*, Imaging Quality and Subject Consistency), while also achieving the highest speedups (1.3× on Self Forcing and 1.19× on LongLive). Notably, LIGHT FORCING yields a higher Total Score than dense FlashAttention (Dao, 2023) baselines (84.5 *vs.* 84.1 for Self Forcing and 83.9 *vs.* 83.2 for LongLive), suggesting substantial redundancy in dense attention and indicating that properly designed sparse solutions can achieve lossless performance.

Moreover, LIGHT FORCING demonstrates strong versatility and applicability, primarily in three aspects compared to prior methods. ① *Sparsity ratio.* Even with the

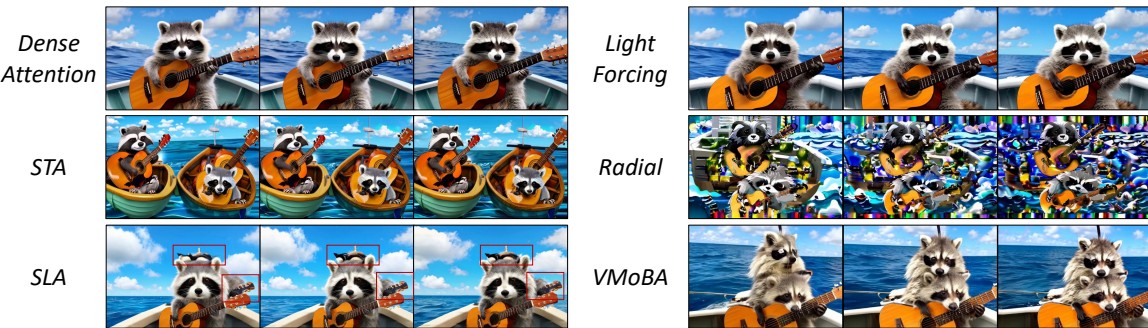

Figure 5. Qualitative comparisons of 5-second videos generated under the prompt "A cute raccoon playing guitar in a boat on the ocean" on Self Forcing (Huang et al., 2025a). We select frames at 0s, 2s, and 5s as representative snapshots of the video.

smallest window size (*i.e.*, $(3, 3, 3)$), STA (Zhang et al., 2025d) attains only 62.5% sparsity under such output resolution, and thus yields limited speedup. ② *Extra overhead.* While permutation-based methods such as SVG2 (Yang et al., 2025b) achieve relatively strong performance among training-free methods, they require repeated clustering initialization as the KV cache evolves, incurring particularly large extra overhead for few-step generators. ③ *Training difficulty.* LongLive (Yang et al., 2025a) adopts LoRA (Hu et al., 2022) to finetune, making it challenging for some chunk-agnostic finetunable methods (*e.g.*, VMoBA (Wu et al., 2025a)) to converge. Even after fine-tuning, their performance still remains far from satisfactory.

Qualitative comparisons are shown in Fig. 5. This further highlights that LIGHT FORCING preserves high-fidelity and consistent video examples, whereas other baselines exhibit pronounced degradation, including object duplication in multi-object scenes (*e.g.*, STA (Zhang et al., 2025d) and VMoBA (Wu et al., 2025a) producing two or more raccoons), anomalous objects (*e.g.*, SLA (Zhang et al., 2025a) generating multiple handles of the guitar), and severe color shifts and artifacts (*e.g.*, Radial (Li et al., 2025c)). These observations suggest that LIGHT FORCING better mitigates error accumulation and over-exposure effects, enabling high-quality long-duration video synthesis.

### 5.3. Longer Generation

For longer videos, since Self Forcing (Huang et al., 2025a) does not natively support generation beyond 5 seconds, we evaluate LIGHT FORCING with Infinite-Forcing (Junyi Chen, 2025) on 15-second videos using VBench-Long (Huang et al., 2025c). As shown in Tab. 2, LIGHT FORCING still maintains a competitive and slightly higher Total Score than Infinite-Forcing (84.1 *vs.* 83.6), suggesting that the proposed LIGHT FORCING remains stable under longer autoregressive rollouts. In particular, LIGHT FORCING improves Quality Score from 84.6 to 85.4, while achieving better Imaging Quality (69.5 *vs.* 68.7), Motion

Smoothness (98.6 *vs.* 98.5), and Dynamic Degree (64.7 *vs.* 54.7). The qualitative long-video results in Fig. 6 further verify this observation, showing that LIGHT FORCING preserves favorable motion smoothness and visual quality under 15-second generation. Complete VBench-Long results are provided in the Appendix.

### 5.4. Ablation Studies

**Ablation for Components.** We evaluate the effect of our two components on Self Forcing (Huang et al., 2025a). We observe that directly applying 1D sparse attention (90% sparsity) without fine-tuning results in a severe quality collapse (*e.g.*, degraded visual fidelity and dynamics). Although further fine-tuning partially recovers performance, it still falls short of dense attention (84.1 *vs.* 82.8 in Total Score). When combined with CAG, the model exhibits notable gains in Aesthetic Quality and Imaging Quality, but its dynamics deteriorate, suggesting that under aggressive sparsity the model relies more heavily on priors from preceding chunks at the expense of motion. In contrast, introducing HSA substantially improves dynamics and ultimately surpasses dense attention in Total Score. To more intuitively demonstrate the role of each component, we also include long-video visualizations of CAG and HSA in the Appendix.

**Sensitivity Analysis on HSA.** We conduct a hyperparameter study on the number of retrieved frames ($topk$) in the first-stage retrieval of HSA, reported in Tab. 4. Due to limited time and resources, we evaluate three settings ($topk \in \{6, 9, 12\}$). Our method remains highly robust, achieving similarly strong performance across these choices. This further suggests that, for each query block, attending to only a small subset of past frames is sufficient to mitigate inconsistency issues.

### 5.5. Efficient Deployment

To better unlock the acceleration potential of autoregressive video generation models, we deploy LIGHT FORC-ING on the mainstream video generation inference frame-

*Table 2.* Long-video generation results on VBench-Long (Huang et al., 2025c). We report representative metrics and the final scores for 15-second videos.

| Method | Aesthetic Quality ↑ | Imaging Quality ↑ | Motion Smoothness ↑ | Dynamic Degree ↑ | Subject Consistency ↑ | Background Consistency ↑ | Quality Score ↑ | Semantic Score ↑ | Total Score ↑ |
|---|---|---|---|---|---|---|---|---|---|
| Infinite-Forcing 1.3B (`fps = 16`) | | | | | | | | | |
| FlashAttention2 (Dao, 2023) | 65.0 | 68.7 | 98.5 | 54.7 | 98.3 | 97.6 | 84.6 | 79.7 | 83.6 |
| LIGHT FORCING | 65.1 | 69.5 | 98.6 | 64.7 | 98.0 | 97.0 | 85.4 | 79.3 | 84.1 |

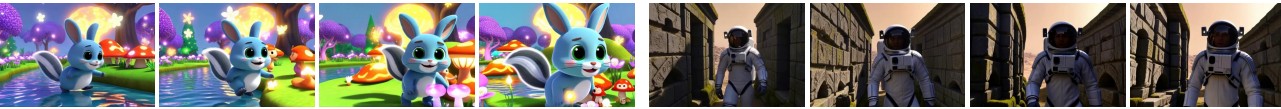

*Figure 6.* Qualitative examples of 15-second videos generated using LIGHT FORCING. Detailed prompts are provided in the Appendix.

*Table 3.* Ablation results for each component of LIGHT FORC- ING. "+1D Sparse Attention" means directly applying sparse attention (Zhang et al., 2025b) under the pretrained Self Forcing weights (Huang et al., 2025a).

| Method | Subject Consistency ↑ | Aesthetic Quality ↑ | Imaging Quality ↑ | Dynamic Degree ↑ | Total Score ↑ |
|---|---|---|---|---|---|
| FLASH ATTENTION | 95.3 | 67.4 | 70.0 | 63.1 | 84.1 |
| *+1D Sparse Attention* | 86.9 | 51.4 | 66.0 | 52.8 | 73.0 |
| *+Finetune* | 94.9 | 65.1 | 69.8 | 46.4 | 82.8 |
| + CAG | 96.1 | 67.7 | 71.0 | 37.5 | 83.2 |
| + CAG & HSA | 96.2 | 67.2 | 71.0 | 66.7 | **84.5** |

*Table 4.* Ablation study on the number of retrieved frames ($topk$) in Hierarchical Sparse Attention (HSA).

| Top-k | Quality Score ↑ | Semantic Score ↑ | Total Score ↑ |
|---|---|---|---|
| 6 | **85.4** | **80.9** | **84.5** |
| 9 | 85.2 | 80.8 | 84.4 |
| 12 | 85.1 | **80.9** | 84.3 |

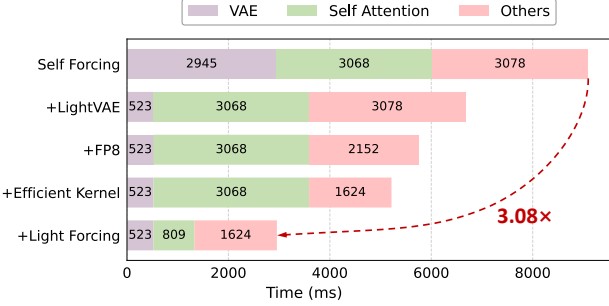

*Figure 7.* Efficient deployment of Light Forcing. We measure its latency on RTX 5090 for 5-second video generation.

work LightX2V (Contributors, 2025) and profile the run-time latency of each component (see Fig. 7). We replace the default Wan VAE (Wan et al., 2025) with the efficient LightVAE (Contributors, 2025), and further deploy all linear layers in the model using low-bit FP8 precision, where weights are quantized with per-channel granularity and activations with per-token granularity. Both choices are widely adopted as nearly lossless acceleration techniques. In addition, we provide several Triton-based optimized kernels, including RoPE, RMSNorm, and `fuse_scale_shift`, to further reduce operator overhead in the end-to-end inference pipeline. In our latency evaluation, measured with our open-source Docker image[5], LIGHT FORCING achieves a $3.79\times$ speedup in attention time and a $3.08\times$ end-to-end speedup, while maintaining satisfactory generation quality. Remarkably, LIGHT FORCING 1.3B achieves 27.4 FPS, enabling real-time video generation on a consumer-grade GPU for the first time. Additional latency evaluations on other GPU platforms (*e.g.*, H100 and A100) and longer durations are provided in the Appendix.

## 6. Conclusion

We proposed LIGHT FORCING, a sparse attention framework tailored for autoregressive video diffusion. By introducing chunk-aware growth and hierarchical sparse attention, our method effectively mitigates error accumulation while preserving long-range context. Extensive experiments demonstrate consistent improvements in both efficiency and generation quality, enabling real-time video synthesis on consumer GPUs and establishing a strong foundation for scalable AR video generation.

## Acknowledgements

This research/project is supported by A*STAR under the RIE2025 Industry Alignment Fund – Industry Collaboration Projects (IAF-ICP) Funding Initiative (Award: I2501E0045), as well as cash and in-kind contribution from the industry partner(s). This research/project is also supported by the NTU Start-Up Grant, Singapore.

---

[5] `lvchengtao/light_forcing:v1`.

## Impact Statement

This paper presents work whose goal is to advance the field of Machine Learning. There are many potential societal consequences of our work, none of which we feel must be specifically highlighted here.

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

## A. Implementation details of Baselines

Since most existing sparse attention methods are originally designed for bidirectional video generation models, applying them to autoregressive video generation requires additional clarification and careful consideration.

- STA (Zhang et al., 2025d): STA partitions tokens into 3D tiles and applies sparse attention to neighboring tiles. In all experiments, we use a window size of $(3, 3, 3)$. The original paper keeps early timesteps in dense attention. Since autoregressive models are typically few-step generators (*e.g.*, 4 steps), we do not adopt this setting and apply sparse attention to all steps.

- Radial Attention (Li et al., 2025c): Since the key-value (KV) sequence length varies over chunks in autoregressive video generation, the effective sparsity ratio of Radial Attention also changes accordingly. For 5 s videos, we perform inference over 7 chunks (3 frames per chunk) with the following sparsity ratios: 67.7, 76.9, 80.6, 82.4, 83.5, 84.9, 86.5.

- SVG2 (Yang et al., 2025b): Since SVG2 relies on K-means clustering and the sequence length in autoregressive generators is shorter than that in bidirectional models, we adjust several hyperparameters to improve its runtime efficiency: `num_q_centroids`=50, `num_k_centroids`=100, `kmeans_iter_init`=20, `top_p_kmeans`=0.9, `min_kc_ratio`=0.10, and `kmeans_iter_step`=2. Notably, whenever the KV length changes in AR models, we re-initialize K-means clustering accordingly.

- LIGHT FORCING: 1) We provide detailed duration-specific settings for both short- and long-video generation in Tab. 5, including the target sparsity ratio, $n_{\text{past\_keep}}$ (the number of retained past frames), $n_{\text{sink}}$ (the number of retained earliest historical frames), $n_{\text{win}}$ (the number of retained nearest historical frames), and the corresponding open-source model links. 2) HSA is activated only when the number of historical frames is larger than $n_{\text{past\_keep}}$. Otherwise, all historical frames are preserved. 3) As discussed in the main paper, we keep dense attention for the first chunk and apply Chunk-Aware Growth for sparsity allocation in subsequent chunks. This is because the first chunk has a relatively short sequence length, so sparse attention yields limited speedup but can cause pronounced performance degradation.

*Table 5.* Hyperparameter settings of LIGHT FORCING for 5-second and 15-second video generation.

| Duration | $s_{\text{target}}$ | $s_{\text{base}}$ | $n_{\text{past\_keep}}$ | $n_{\text{sink}}$ | $n_{\text{win}}$ | Model |
|---|---|---|---|---|---|---|
| 5s | 0.88 | 0.98 | 6 | 1 | 2 | link |
| 15s | 0.85 | 0.95 | 3 | 1 | 1 | link |

## B. Prompts for Long Video Generation

The prompts used for the qualitative examples of 15-second long video generation in the main paper are listed below.

**Prompt 1.** A 3D animation of a small, round, fluffy creature with big, expressive eyes exploring a vibrant, enchanted forest. The creature, a whimsical blend of a rabbit and a squirrel, has soft blue fur and a bushy, striped tail. It hops along a sparkling stream, its eyes wide with wonder. The forest is alive with magical elements: flowers that glow and change colors, trees with leaves in shades of purple and silver, and small floating lights that resemble fireflies. The creature stops to interact playfully with a group of tiny, fairy-like beings dancing around a mushroom ring. The creature looks up in awe at a large, glowing tree that seems to be the heart of the forest. The scene is rendered in a detailed, fantasy style, with a soft, ethereal lighting that enhances the enchantment. The camera follows the creature as it moves, capturing its playful interactions and the magical ambiance of the forest. A medium shot with a dynamic angle that highlights the creature's expressions and the enchanting environment.

**Prompt 2.** An astronaut in a sleek, white spacesuit walks between two ancient stone buildings, their surfaces adorned with intricate carvings and moss. The astronaut's helmet reflects the dim, otherworldly light casting shadows across the worn stones. The buildings loom large, creating a narrow passage that seems to stretch into the distance. The background shows a barren landscape with distant, rocky hills and a pale, orange sky. The astronaut moves with a determined gait, one hand on the building's surface, the other holding a small device. The photo has a realistic, high-resolution texture, capturing the astronaut's focused expression and the textures of the ancient architecture. A medium shot from a slightly elevated angle, emphasizing the contrast between the modern astronaut and the ancient structures.

## C. Theoretical proof of CAG

**Denoising-with-re-noising Markov kernel (chunk-wise).** Fix an AR chunk index $i$ and conditioning $(\boldsymbol{x}^{<i}, c)$, and let the inference schedule be $t_T > t_{T-1} > \cdots > t_0$ with corresponding noise levels $\{\sigma_{t_j}\}_{j=0}^{T} \subset (0, 1]$. The transition operator $\Psi$ induces the stochastic update

$$\boldsymbol{x}_{t_{j-1}}^i = \Psi\Big(G_\theta(\boldsymbol{x}_{t_j}^i, t_j, \boldsymbol{x}^{<i}, c), \boldsymbol{\epsilon}_{t_{j-1}}, t_{j-1}\Big) = (1 - \sigma_{t_{j-1}}) \, G_\theta(\boldsymbol{x}_{t_j}^i, t_j, \boldsymbol{x}^{<i}, c) + \sigma_{t_{j-1}} \, \boldsymbol{\epsilon}_{t_{j-1}}, \tag{14}$$

where $\boldsymbol{\epsilon}_{t_{j-1}} \sim \mathcal{N}(\mathbf{0}, \mathbf{I})$ are i.i.d. across steps. Hence, conditional on $\boldsymbol{x}_{t_j}^i$, the transition is Gaussian:

$$\boldsymbol{x}_{t_{j-1}}^i \mid \boldsymbol{x}_{t_j}^i \sim \mathcal{N}\big(\boldsymbol{\mu}_{\theta,j}(\boldsymbol{x}_{t_j}^i), \, \sigma_{t_{j-1}}^2 \mathbf{I}\big), \qquad \boldsymbol{\mu}_{\theta,j}(y) := (1 - \sigma_{t_{j-1}}) \, G_\theta(y, t_j, \boldsymbol{x}^{<i}, c). \tag{15}$$

**Ideal reverse kernel and mean-map error.** Let $\boldsymbol{\mu}_j^\star(\cdot)$ be the *ideal* reverse mean map (defined by the exact score / optimal denoiser under the same schedule), and denote by $p_0(\cdot \mid \boldsymbol{x}^{<i}, c)$ the true conditional data distribution of $\boldsymbol{x}^i$. Let $q_0(\cdot \mid \boldsymbol{x}^{<i}, c)$ be the distribution of the generated output after $T$ transitions from $\boldsymbol{x}_{t_T}^i \sim \mathcal{N}(\mathbf{0}, \mathbf{I})$. Assume the (average) conditional mean-map error

$$\frac{1}{T} \sum_{j=1}^{T} \mathbb{E}\Big[\big\|\boldsymbol{\mu}_{\theta,j}(\boldsymbol{x}_{t_j}^i) - \boldsymbol{\mu}_j^\star(\boldsymbol{x}_{t_j}^i)\big\|_2^2 \;\Big|\; \boldsymbol{x}^{<i}, c\Big] \;\leq\; \varepsilon_{\text{mean}}^2, \tag{16}$$

which is implied by the score/denoiser estimation accuracy as in the assumptions of Theorem 3 in (Li et al., 2023).

**Stepwise KL control (Gaussian KL).** For Gaussians with equal covariance, we have

$$\mathrm{KL}(\mathcal{N}(\boldsymbol{\mu}^\star, \Sigma) \,\|\, \mathcal{N}(\boldsymbol{\mu}, \Sigma)) = \frac{1}{2} \big\|\Sigma^{-1/2}(\boldsymbol{\mu}^\star - \boldsymbol{\mu})\big\|_2^2. \tag{17}$$

Using $\Sigma = \sigma_{t_{j-1}}^2 \mathbf{I}$ in equation 15 yields the per-step contribution

$$\mathbb{E}\Big[\mathrm{KL}\Big(\mathcal{N}(\boldsymbol{\mu}_j^\star(\boldsymbol{x}_{t_j}^i), \sigma_{t_{j-1}}^2 \mathbf{I}) \,\Big\|\, \mathcal{N}(\boldsymbol{\mu}_{\theta,j}(\boldsymbol{x}_{t_j}^i), \sigma_{t_{j-1}}^2 \mathbf{I})\Big) \;\Big|\; \boldsymbol{x}^{<i}, c\Big] = \frac{1}{2\sigma_{t_{j-1}}^2} \mathbb{E}\Big[\big\|\boldsymbol{\mu}_{\theta,j}(\boldsymbol{x}_{t_j}^i) - \boldsymbol{\mu}_j^\star(\boldsymbol{x}_{t_j}^i)\big\|_2^2 \;\Big|\; \boldsymbol{x}^{<i}, c\Big]. \tag{18}$$

**Telescoping KL and TV bound.** Applying the KL chain rule along the Markov chain induced by equation 14, one obtains

$$\mathrm{KL}\big(q_0(\cdot \mid \boldsymbol{x}^{<i}, c) \,\|\, p_0(\cdot \mid \boldsymbol{x}^{<i}, c)\big) \;\leq\; \sum_{j=1}^{T} \frac{1}{2\sigma_{t_{j-1}}^2} \mathbb{E}\Big[\big\|\boldsymbol{\mu}_{\theta,j}(\boldsymbol{x}_{t_j}^i) - \boldsymbol{\mu}_j^\star(\boldsymbol{x}_{t_j}^i)\big\|_2^2 \;\Big|\; \boldsymbol{x}^{<i}, c\Big] \;+\; \mathrm{KL}(q_{t_T} \,\|\, p_{t_T}). \tag{19}$$

By Pinsker's inequality,

$$\mathrm{TV}\big(q_0(\cdot \mid \boldsymbol{x}^{<i}, c), \, p_0(\cdot \mid \boldsymbol{x}^{<i}, c)\big) \leq \sqrt{\frac{1}{2} \mathrm{KL}(q_0(\cdot \mid \boldsymbol{x}^{<i}, c) \,\|\, p_0(\cdot \mid \boldsymbol{x}^{<i}, c))}. \tag{20}$$

**Error vs. number of denoising steps $T$.** Under the same regularity and schedule conditions in (Li et al., 2023), the discretization/mixing term contributes $\tilde{\mathcal{O}}(d/T)$ in KL (polylog factors), while the model (score) error contributes $\tilde{\mathcal{O}}(d\,\varepsilon_{\text{score}}^2)$ in KL. Combining equation 20 with Eq. (31) of (Li et al., 2023) gives the conditional TV guarantee

$$\mathrm{TV}\big(q_0(\cdot \mid \boldsymbol{x}^{<i}, c), \, p_0(\cdot \mid \boldsymbol{x}^{<i}, c)\big) \;\leq\; C_1 \cdot \frac{d \log^3 T}{\sqrt{T}} \;+\; C_2 \cdot \sqrt{d}\,\varepsilon_{\text{score}} \log^2 T, \tag{21}$$

for universal constants $C_1, C_2 > 0$. In particular, when $\varepsilon_{\text{score}} = 0$, the sampling error decays as $\tilde{\mathcal{O}}(d/\sqrt{T})$, whereas for imperfect models it saturates at $\tilde{\mathcal{O}}(\sqrt{d}\,\varepsilon_{\text{score}})$ as $T \to \infty$.

## D. Detailed VBench Results

We report the full VBench (Huang et al., 2024b) results across all dimensions for each method in Tab. 6 and Tab. 7.

*Table 6.* Detailed performance comparison with state-of-the-art baselines on VBench (Huang et al., 2024b) (quality part).

| Method | Subject Consistency↑ | Background Consistency↑ | Temporal Flickering↑ | Motion Smoothness↑ | Aesthetic Quality↑ | Imaging Quality↑ | Dynamic Degree↑ |
|---|---|---|---|---|---|---|---|
| Self-Forcing 1.3B (`fps = 16`) | | | | | | | |
| FlashAttention2 (Dao, 2023) | 95.3 | 96.5 | 99.1 | 98.3 | 67.4 | 70.0 | 63.1 |
| STA (Zhang et al., 2025d) | 96.3 | 96.9 | 99.2 | 98.5 | 64.5 | 71.7 | 48.9 |
| Radial (Li et al., 2025c) | 90.2 | 93.6 | 95.6 | 96.0 | 45.8 | 66.1 | 88.6 |
| SVG2 (Yang et al., 2025b) | 93.6 | 95.6 | 98.2 | 97.8 | 66.0 | 68.2 | 72.8 |
| SLA (Zhang et al., 2025a) | 95.6 | 96.7 | 99.2 | 98.3 | 66.7 | 69.8 | 44.2 |
| VMoBA (Wu et al., 2025a) | 92.8 | 95.5 | 98.0 | 97.3 | 65.2 | 69.9 | 84.2 |
| LIGHT FORCING | **96.2** | 96.5 | **99.2** | 98.3 | **67.2** | **71.0** | 66.7 |
| LongLive 1.3B (`fps = 16`) | | | | | | | |
| FlashAttention2 (Dao, 2023) | 97.0 | 97.2 | 99.3 | 98.8 | 68.7 | 69.3 | 39.2 |
| STA (Zhang et al., 2025d) | 97.4 | 97.8 | 99.6 | 99.0 | 65.6 | 71.2 | 22.8 |
| Radial (Li et al., 2025c) | 77.6 | 88.9 | 98.1 | 98.0 | 55.1 | 72.0 | 25.0 |
| SVG2 (Yang et al., 2025b) | 95.3 | 96.1 | 98.8 | 98.5 | 66.7 | 67.0 | 44.4 |
| VMoBA (Wu et al., 2025a) | 58.3 | 80.9 | 97.6 | 97.5 | 59.9 | 68.2 | 50.6 |
| LIGHT FORCING | **96.9** | 96.7 | 98.9 | 98.2 | **67.2** | 70.6 | **59.4** |

*Table 7.* Detailed performance comparison with state-of-the-art baselines on VBench (Huang et al., 2024b) (semantic part).

| Method | Object Class↑ | Multiple Objects↑ | Human Action↑ | Color↑ | Spatial Relationship↑ | Scene↑ | Appearance Style↑ | Temporal Style↑ | Overall Consistency↑ |
|---|---|---|---|---|---|---|---|---|---|
| Self-Forcing 1.3B (`fps = 16`) | | | | | | | | | |
| FlashAttention2 (Dao, 2023) | 94.9 | 88.4 | 96.4 | 88.6 | 83.1 | 54.4 | 20.6 | 24.6 | 26.9 |
| STA (Zhang et al., 2025d) | 95.2 | 86.1 | 95.2 | 91.7 | 91.1 | 57.1 | 22.1 | 23.0 | 25.5 |
| Radial (Li et al., 2025c) | 56.0 | 31.8 | 80.4 | 86.5 | 39.0 | 15.1 | 22.3 | 15.9 | 18.1 |
| SVG2 (Yang et al., 2025b) | 93.5 | 73.5 | 96.4 | 87.8 | 76.9 | 54.2 | 20.4 | 24.5 | 27.0 |
| SLA (Zhang et al., 2025a) | 96.4 | 87.5 | 96.8 | 91.8 | 89.3 | 56.3 | 20.5 | 24.2 | 26.8 |
| VMoBA (Wu et al., 2025a) | 93.9 | 81.1 | 96.8 | 90.5 | 81.0 | 55.3 | 20.5 | 24.3 | 26.8 |
| LIGHT FORCING | 94.3 | **88.9** | 96.0 | 88.2 | 81.4 | 55.3 | 20.1 | 24.6 | **26.9** |
| LongLive 1.3B (`fps = 16`) | | | | | | | | | |
| FlashAttention2 (Dao, 2023) | 94.4 | 87.8 | 96.4 | 89.2 | 78.5 | 55.6 | 20.6 | 24.3 | 26.7 |
| STA (Zhang et al., 2025d) | 95.5 | 86.0 | 95.4 | 94.7 | 90.4 | 54.8 | 21.7 | 22.2 | 25.1 |
| Radial (Li et al., 2025c) | 72.7 | 59.9 | 93.2 | 93.0 | 66.6 | 16.7 | 22.4 | 19.4 | 22.4 |
| SVG2 (Yang et al., 2025b) | 92.3 | 78.9 | 96.0 | 88.5 | 74.5 | 53.9 | 20.4 | 24.4 | 27.0 |
| VMoBA (Wu et al., 2025a) | 78.2 | 64.8 | 96.4 | 89.3 | 64.6 | 26.8 | 21.5 | 23.2 | 26.0 |
| LIGHT FORCING | **95.3** | **89.6** | 96.6 | 86.1 | 78.7 | 53.3 | 20.1 | 24.4 | **26.7** |

*Table 8.* Comparison with feature-cache and token-reduction acceleration methods on latency and representative VBench (Huang et al., 2024b) metrics.

| Method | Latency (s)↓ | Speedup↑ | Subject Consistency↑ | Background Consistency↑ | Aesthetic Quality↑ | Imaging Quality↑ | Multiple Objects↑ | Scene↑ |
|---|---|---|---|---|---|---|---|---|
| TeaCache (Liu et al., 2025a) | 8.32 | 1.15× | 94.9 | 96.1 | 67.0 | 69.5 | 84.4 | 52.2 |
| DyCoke (Tao et al., 2025) | 8.71 | 1.10× | 94.5 | 95.6 | 67.2 | 69.5 | 87.5 | 54.2 |
| LIGHT FORCING | **7.39** | **1.30×** | **96.2** | **96.5** | **67.2** | **71.0** | **88.9** | **55.3** |

# E. Detailed VBench-Long Results

We report the full VBench-Long (Huang et al., 2025c) results for 15-second generation in Tab. 9.

# F. Qualitative Ablation Study

We further provide long-video qualitative ablations for the two proposed modules in LIGHT FORCING, *i.e.*, Chunk-Aware Growth (CAG) and Hierarchical Sparse Attention (HSA). All three variants use the trained long-video version of LIGHT

*Table 9.* Detailed long-video generation comparison on VBench-Long (Huang et al., 2025c).

| Method | Subject Consistency↑ | Background Consistency↑ | Temporal Flickering↑ | Motion Smoothness↑ | Aesthetic Quality↑ | Imaging Quality↑ | Dynamic Degree↑ | Object Class↑ |
|---|---|---|---|---|---|---|---|---|
| *Infinite-Forcing 1.3B (`fps = 16`)* | | | | | | | | |
| FlashAttention2 (Dao, 2023) | 98.3 | 97.6 | 99.5 | 98.5 | 65.0 | 68.7 | 54.7 | 95.1 |
| LIGHT FORCING | 98.0 | 97.0 | 99.4 | 98.6 | 65.1 | 69.5 | 64.7 | 95.4 |

| Method | Multiple Objects↑ | Human Action↑ | Color↑ | Spatial Relationship↑ | Scene↑ | Appearance Style↑ | Temporal Style↑ | Overall Consistency↑ |
|---|---|---|---|---|---|---|---|---|
| *Infinite-Forcing 1.3B (`fps = 16`)* | | | | | | | | |
| FlashAttention2 (Dao, 2023) | 84.7 | 96.1 | 86.2 | 74.0 | 57.8 | 20.7 | 24.1 | 26.1 |
| LIGHT FORCING | 85.1 | 94.5 | 85.3 | 77.5 | 52.7 | 20.5 | 24.6 | 26.2 |

FORCING[6]. For the naive baseline, we adopt a fixed sparsity ratio of 85% across chunks. As shown in Fig. 8 and Fig. 9, the naive strategy suffers from severe color degradation, and the first example exhibits noticeable noise starting from around 5s. Adding CAG largely alleviates the color degradation by allocating the sparsity budget according to chunk-wise generation difficulty, but white band-like artifacts still appear in the latter part of the videos (*e.g.*, around 10s). After further incorporating HSA, these artifacts are effectively mitigated, leading to more stable long-video generation with better visual quality and temporal consistency.

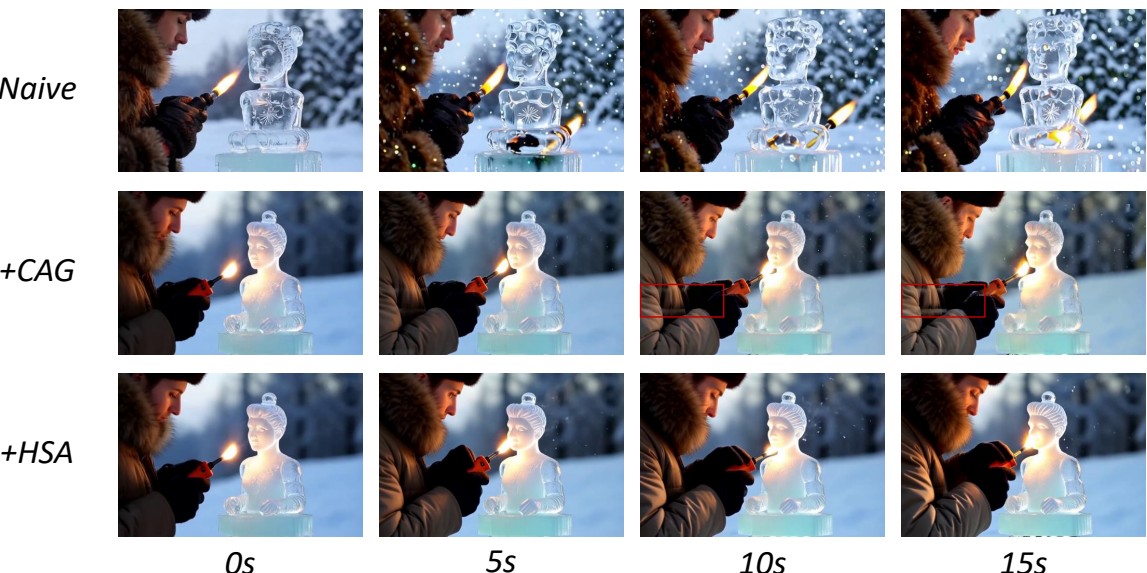

*Figure 8.* Qualitative ablation study of CAG and HSA for long-video generation on the first example.

## G. Comparison with Other Acceleration Methods

We further compare LIGHT FORCING with two representative acceleration paradigms beyond sparse attention: feature caching and token reduction. TeaCache (Liu et al., 2025a) estimates the change between model outputs from timestep embeddings and uses this signal to decide when intermediate model outputs can be reused during denoising. DyCoke (Tao et al., 2025) combines temporal token merging with dynamic KV-cache pruning, aiming to remove redundant tokens while preserving visually important information.

For implementation, we set the TeaCache threshold to 0.2. For DyCoke, we follow the LLMC+ (Lv et al., 2026) implementation and adapt it to the chunk-wise generation protocol of Self Forcing (Huang et al., 2025a). Specifically, we group three consecutive frames as one unit, matching the three-frame chunk used by Self Forcing. Within each group, we keep all

---

[6]https://huggingface.co/mack-williams/Light-Forcing/blob/main/long_video_gen.pt

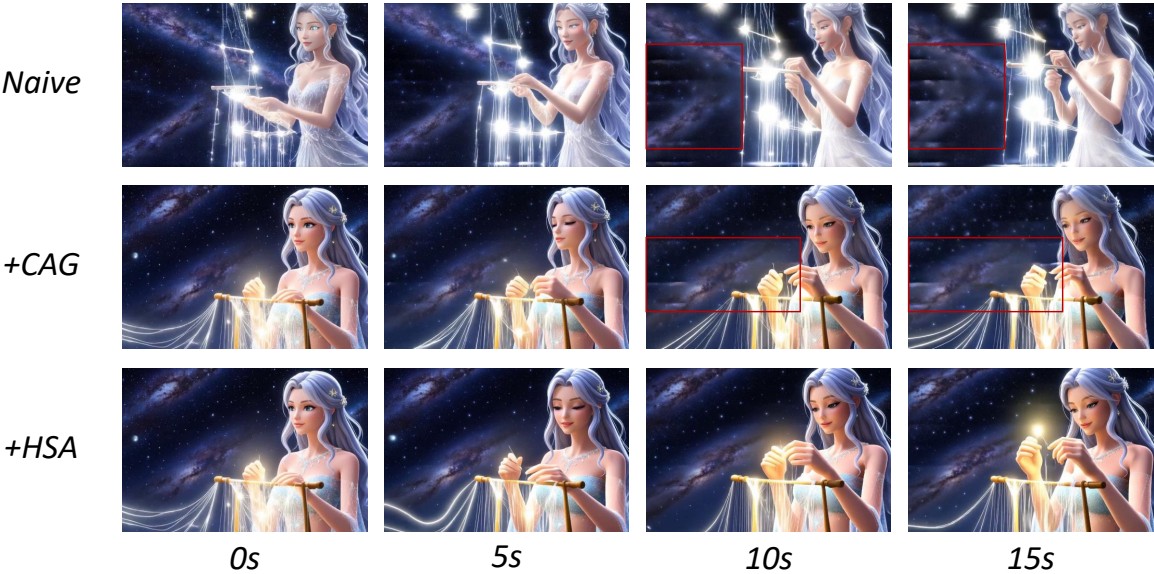

*Figure 9.* Qualitative ablation study of CAG and HSA for long-video generation on the second example.

tokens from the first frame and retain only $25\%$ of the tokens from the second and third frames, resulting in a total historical KV-cache retention ratio of $50\%$.

As shown in Tab. 8, under the highest-speedup setting among these methods ($1.30\times$), LIGHT FORCING achieves the best accuracy on all reported representative metrics. It obtains higher subject consistency ($96.2$ vs. $94.9/94.5$), background consistency ($96.5$ vs. $96.1/95.6$), imaging quality ($71.0$ vs. $69.5/69.5$), multiple-object reasoning ($88.9$ vs. $84.4/87.5$), and scene understanding ($55.3$ vs. $52.2/54.2$) than TeaCache and DyCoke, respectively. These results suggest that directly exploiting the autoregressive attention structure preserves visual fidelity more effectively than reusing intermediate features or aggressively reducing historical tokens. We also observe in qualitative comparisons that TeaCache tends to introduce blurry frames, while DyCoke often causes abrupt motion changes due to temporal token removal.

## H. More Efficiency Analysis

### H.1. More Devices

To further assess the hardware generality of LIGHT FORCING, we evaluate the same inference pipeline on three representative GPU platforms: RTX 5090, A100, and H100. All measurements use a single GPU, record the generation time of one video after operator warm-up, and start timing from the second generated sample. Unless otherwise specified, all experiments are conducted with the same Docker image[7], which provides the optimized sparse-attention, FP8, RoPE, RMSNorm, and related deployment kernels used in our implementation. On RTX 5090 and A100, LIGHT FORCING invokes the Triton sparse-attention kernel, while on H100 it uses the FlashAttention 4 (Zadouri et al., 2026) sparse kernel.

As shown in Tab. 10, LIGHT FORCING consistently improves end-to-end latency across consumer and datacenter GPUs. On RTX 5090, sparse attention alone yields $1.33\times$ and $1.26\times$ acceleration for 5s and 15s generation, respectively, and the complete deployment stack reaches $3.07\times$–$3.17\times$ speedup. On H100, although attention computation for the 1.3B model is already relatively lightweight on H-series GPUs, LIGHT FORCING still provides $1.11\times$–$1.12\times$ speedup by itself and nearly $2\times$ acceleration when combined with the remaining deployment optimizations. On A100, where FP8 linear layers are not supported and the sparse kernel has not been specifically tuned for this architecture, LIGHT FORCING still achieves positive gains and reaches $2.12\times$–$2.35\times$ speedup with efficient kernel fusion and LightVAE. These results indicate that LIGHT FORCING is not tied to a single accelerator or kernel implementation, but provides broadly applicable sparse-attention acceleration for autoregressive video diffusion.

---

[7]`lvchengtao/light_forcing:v1`.

*Table 10.* Additional efficiency results across different devices. "Dense" denotes the dense-attention baseline on the corresponding device. Each cell reports latency and the end-to-end speedup over the dense baseline under the same duration and device setting.

| Device | Duration | Dense | +LIGHT FORCING | +FP8 Linear | +Efficient Kernel | +LightVAE | Final Speedup |
|--------|----------|-------|----------------|-------------|-------------------|-----------|---------------|
| RTX 5090 | 5s | 9.09s (1.00×) | 6.83s (1.33×) | 5.90s (1.54×) | 5.37s (1.69×) | 2.96s (3.07×) | 3.07× |
| RTX 5090 | 15s | 30.40s (1.00×) | 24.20s (1.26×) | 21.40s (1.42×) | 17.00s (1.79×) | 9.60s (3.17×) | 3.17× |
| A100 | 5s | 11.38s (1.00×) | 9.88s (1.15×) | – | 9.41s (1.21×) | 4.85s (2.35×) | 2.35× |
| A100 | 15s | 38.28s (1.00×) | 34.56s (1.11×) | – | 26.63s (1.44×) | 18.08s (2.12×) | 2.12× |
| H100 | 5s | 4.80s (1.00×) | 4.33s (1.11×) | 4.32s (1.11×) | 3.74s (1.28×) | 2.39s (2.01×) | 2.01× |
| H100 | 15s | 15.80s (1.00×) | 14.10s (1.12×) | 13.80s (1.14×) | 12.10s (1.31×) | 8.00s (1.98×) | 1.98× |

## H.2. Peak Memory Analysis

We also report peak GPU memory in Tab. 11. The available peak-memory measurements are collected on RTX 5090 under the same Docker image and inference protocol as above.

*Table 11.* Peak memory usage on RTX 5090. Values are measured in GB.

| Duration | Dense | +LIGHT FORCING | +FP8 Linear | +Efficient Kernel | +LightVAE |
|----------|-------|----------------|-------------|-------------------|-----------|
| 5s | 17.8 | 17.8 | 16.6 | 15.8 | 12.7 |
| 15s | 17.6 | 17.6 | 16.5 | 16.3 | 13.1 |

LIGHT FORCING primarily reduces the number of attention tiles evaluated at runtime, so its main benefit is latency reduction rather than peak-memory reduction in the current implementation. Consequently, the measured peak memory remains unchanged after enabling sparse attention alone. FP8 linear layers and efficient kernels reduce the memory footprint by lowering activation or operator overhead, and LightVAE provides the largest memory reduction by replacing the VAE component with a more efficient variant. Overall, the full deployment stack reduces peak memory from 17.8 GB to 12.7 GB for 5s generation and from 17.6 GB to 13.1 GB for 15s generation, corresponding to 28.7% and 25.6% memory reduction, respectively. The memory profile therefore complements the latency results: LIGHT FORCING supplies broadly applicable attention acceleration, while the accompanying deployment modules further improve the feasibility of running autoregressive video diffusion on memory-constrained devices.

## I. More Visualization Examples

We provide more detailed qualitative comparisons in the supplementary material. In Fig. 10, we visualize results for all baselines under two prompts, *"A person is clay pottery making"* and *"Turtle swimming in ocean"*. Most baselines exhibit noticeable degradation, including (1) loss of fine-grained details (*e.g.*, distorted hands) and (2) anomalous generations (*e.g.*, a turtle with two heads).

## J. Limitations

Despite the strong empirical results, our work has several limitations. First, we evaluate only 1.3B models. Scaling LIGHT FORCING to larger models (*e.g.*, a 14B realtime-video model (Millon, 2025)) is an important direction. Second, although HSA can alleviate white band-like artifacts, subtle artifacts may still remain in a very small number of samples, which we leave as a future direction. Third, some of our kernel-fusion operators may provide less pronounced acceleration on certain GPUs, such as A100, since they have not been specifically adapted to these architectures. Finally, while sparsity is effective, combining it with other methods (*e.g.*, reducing generation to very few denoising steps, such as 1–3 steps, or low-bit quantization) to push toward extreme acceleration remains open.

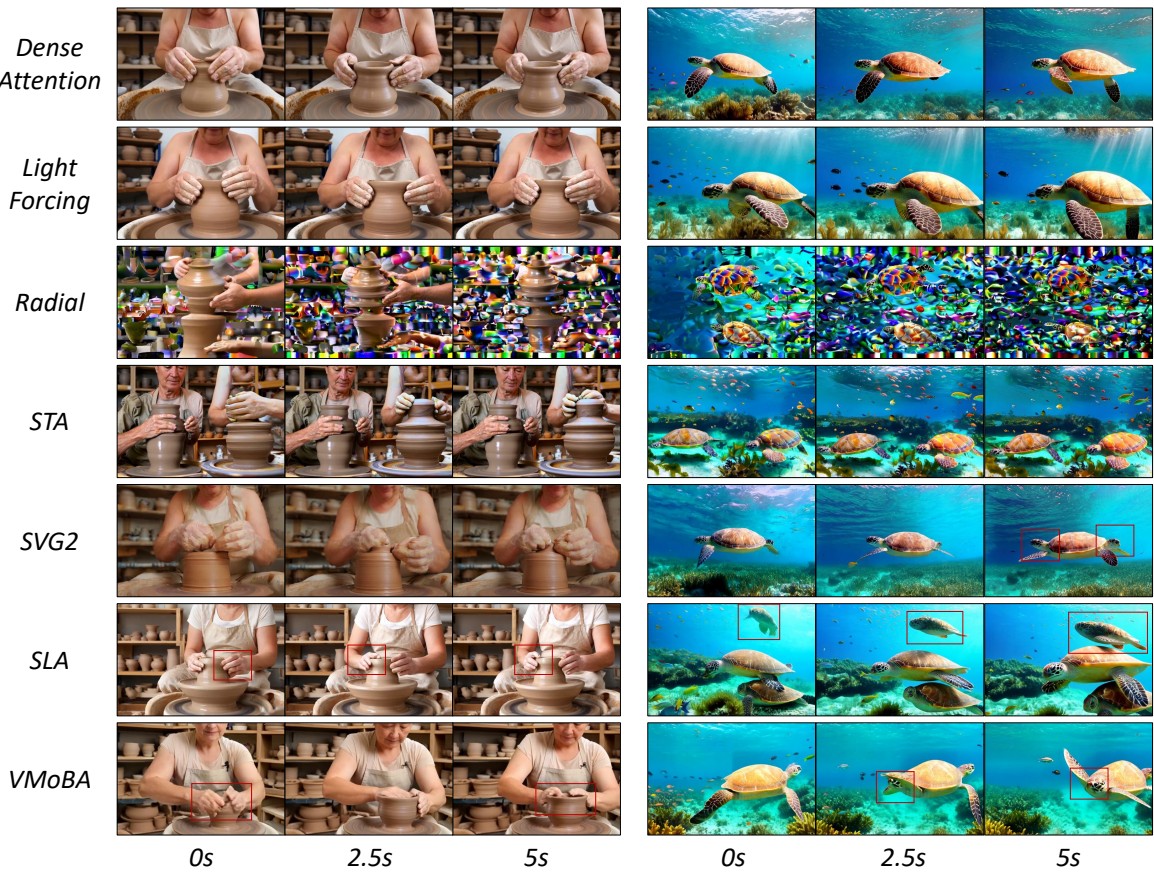

*Figure 10.* More qualitative examples on Self Forcing (Huang et al., 2025a).

