# OpenReview forum: "Light Forcing: Accelerating Autoregressive Video Diffusion via Sparse Attention"
_ICML.cc/2026/Conference — ICML 2026 regular_

### Official Review · Reviewer_3qfq · 2026-02-25

**Soundness:** 2
**Presentation:** 3
**Significance:** 3
**Originality:** 3
**Overall Recommendation:** 4
**Confidence:** 3

**Summary:**

The paper tackles the quadratic cost of attention in autoregressive (AR) video diffusion models, where attention dominates inference time, especially for later chunks. It reports that directly porting sparse-attention methods designed for bidirectional video diffusion significantly harms AR generation quality and attributes this to chunk-wise error accumulation and poor exploitation of historical context. The authors propose LIGHT FORCING, which combines (1) Chunk‑Aware Growth (CAG), a scheme that allocates lower sparsity to early chunks and higher sparsity to later ones based on a bound on total variation distance, and (2) Hierarchical Sparse Attention (HSA), a two-level, coarse‑to‑fine mask selection over frames and blocks. Experiments on Self‑Forcing and LongLive 1.3B models on VBench show higher or comparable quality to dense FlashAttention at 1.19–1.3× end‑to‑end speedup.

**Compliance With Llm Reviewing Policy:**

Affirmed.

**Final Justification:**

This paper addresses an important and timely problem in autoregressive video generation. My initial concerns were mainly about the completeness of the related-work positioning, the reliance on VBench alone for evaluation, the lack of memory/scaling analysis, and the limited validation on only a 1.3B model.

After reading the authors’ rebuttal, I find that several of my main concerns were addressed in a reasonable and constructive manner. Some limitations still remain, especially regarding broader comparisons and more comprehensive evaluation, but I do not view them as fatal weaknesses.

Overall, I think the paper is acceptable, although it is still somewhat borderline, and I would not be surprised if opinions differ.

**Key Questions For Authors:**

1. The paper frames the landscape of AR acceleration primarily in terms of "bidirectional sparse attention methods" but does not discuss other contemporaneous AR acceleration approaches such as those involving temporal cache compression, structured temporal attention, or AR-specific distillation. Could the authors elaborate on how LIGHT FORCING compares to or complements these other approaches, and why these methods were not included in the related work section?
2. All quantitative evaluations rely solely on VBench, without any human evaluation, user preference study, or alternative automatic metrics. Given that VBench is still evolving and may have limitations, how do the authors justify the lack of cross-validation with other metrics or user feedback, and do they plan to evaluate their method using a broader set of metrics in future work?
3. While the paper presents timing improvements from LightVAE, FP8, and LIGHT FORCING, there is no characterization of GPU memory overhead, maximum sequence length supported, or how the speedup scales with growing sequence lengths. Can the authors provide more detailed insights into the memory requirements and performance scaling of their method, especially in the context of large AR generative workloads?
4. The paper evaluates only a 1.3B backbone model, which is mentioned as a limitation in Appendix E. Given the rapid growth in the scale of modern AR video models, would it be possible to include a preliminary investigation of LIGHT FORCING's performance on larger models to identify any scaling trends of CAG/HSA?

Addressing my concerns would lead me to reconsider and potentially raise my score.

**Limitations:**

yes

**Strengths And Weaknesses:**

## Strengths
1. Clear bottleneck diagnosis and AR‑specific framing. Figure 1 very clearly shows attention latency growing linearly with chunk index and dominating total runtime (3.1× versus linear+other by chunk 14). The paper articulates why AR video diffusion has additional sensitivity to sparsity, due to chunk‑wise error compounding and changing KV length, which is genuinely different from bidirectional denoising setups.
2. Technically careful and empirically grounded. Both core components (CAG and HSA) are motivated by concrete observations. Figure 2 reveals the asymmetry between sparsifying early vs late chunks, leading to the idea of chunk‑aware sparsity. Figure 4 visualizes attention maps at various layers/heads and timesteps, showing sink/diagonal structures that sliding windows cannot capture. These diagnostics make the method feel less like a black‑box hack and more like a principled intervention.
3. Good exposition and diagrams. Figures 1, 2, 3, and 6 are particularly useful: they succinctly show the scaling issue, the effect of chunk‑wise sparsity, the architecture of LIGHT FORCING, and the final system speedup. Equation numbering and notation are generally consistent. Section 4 reads well and is accessible even to readers not deeply immersed in AR diffusion.

## Weaknesses
1. Incomplete related‑work positioning on AR acceleration. The paper largely frames the landscape as “bidirectional sparse attention methods that fail when naively transferred to AR.” It omits other contemporaneous AR acceleration approaches that operate via temporal cache compression, structured temporal attention, or AR‑specific distillation. This gap makes it harder to understand how LIGHT FORCING’s contributions compare to or complement those methods.
2. Limited diversity of evaluation metrics beyond VBench. All quantitative evaluations are based on VBench; there is no human evaluation, no user preference study, and no alternative automatic metrics to cross‑validate the results. Given that VBench is still evolving and can have blind spots, this single‑benchmark reliance slightly weakens the generality of the quality claims.
3. Missing memory and scaling characterization. Figure 6 shows timing improvements from LightVAE, FP8, and LIGHT FORCING, but there is no measurement of GPU memory overhead, maximum sequence length supported, or scaling of speedup as sequence length grows. Since AR generative workloads often push memory limits, a more complete systems characterization would significantly strengthen the “practical deployment” argument.
4. Only 1.3B backbones evaluated. The authors mention this limitation in Appendix E. Given that modern AR video models are rapidly growing in scale, it would be valuable to see at least a preliminary investigation at a different model size to check for scaling trends of CAG/HSA.

---

> ### Author Rebuttal · Authors · 2026-03-31
>
> Thanks to the reviewer for the constructive comments.
>
> - **Incomplete baselines among AR acceleration methods.** Thank you for the comment. We would like to clarify that *Self Forcing [1] itself is already an AR-specific distillation method*, which post-trains a bidirectional video diffusion model into a *few-step* autoregressive video diffusion model (4 steps in practice). As a sparse attention technique, Light Forcing is **orthogonal** to such AR-specific distillation methods. In addition, our baselines already include structured temporal attention methods, such as **STA and Radial (Table 1 in paper)**, both of which adopt neighborhood-based attention patterns. We further add two temporal cache compression methods, **TeaCache [2]** and **DyCoke [3]**, with results shown in the table below. Under the **highest** speedup setting (**1.3×**), Light Forcing achieves the **best** accuracy (**84.5** on VBench). We also observe that TeaCache and DyCoke tend to introduce blurry frames and abrupt motion changes, respectively. We believe that current cache-based techniques (*e.g.*, TeaCache), which accelerate generation by skipping certain timesteps, are *not easily* applicable to few-step AR models. We appreciate this suggestion and will add these other contemporaneous AR acceleration approaches in the next version.
>
>
>     |Method|Latency|Quality Score ↑|Semantic Score ↑|Total Score↑|
>     |---|---|---|---|---|
>     |Dense|9.61 (1.0×)|84.8|81.2|84.1|
>     |STA|8.27 (1.16×)|84|82.1|83.6|
>     |Radial|7.39 (1.30×)|78.7|53.7|73.7|
>     |TeaCache|8.32 (1.15×)|84.4|79.9|83.5|
>     |DyCoke|8.71 (1.10×)|83.9|81|83.3|
>     |Light Forcing|7.39 (1.30×)|85.4|80.9|84.5|
>
> [1] Self forcing: Bridging the train-test gap in autoregressive video diffusion. NeurIPS 2025.
>
> [2] Timestep Embedding Tells: It's Time to Cache for Video Diffusion Model. CVPR 2025.
>
> [3] Dycoke: Dynamic compression of tokens for fast video large language models. CVPR 2025.
>
> - **Limited metric diversity**. Thank you for the comment. We further evaluate Light Forcing through a human preference study. Specifically, we use the *10 prompts* from the MovieGenBench dataset and gather ratings from independent evaluators via the Prolific platform. Due to time constraints, we have so far collected preferences from *23 participants*. As shown in the table below, our method outperforms both **SLA and VMoBA** in human preference. We will include more participants and additional baselines in the future version.
>
>
>     ||Light Forcing vs. VMoba|Light Forcing vs. SLA|
>     |---|---|---|
>     |Preference Rate (%)|61.3|67.8|
>
> - **Memory overhead and scaling behavior**. Thank you for the comment on memory overhead. We provide a detailed memory and latency profile across videos of different lengths *without using offloading techniques*. In terms of memory, *based on our estimation*, for 480p generation, the maximum supported sequence length is close to **1 minute**, while for 720p generation the memory cost increases significantly and the supported length is around **20 seconds** on a single 80 GB GPU. In terms of latency, we report both attention latency and DiT latency for a single chunk (*without VAE decoding*). As the token count increases, *e.g.*, with longer generation duration or higher resolution, the speedup becomes more pronounced. For 720p 5-second video generation, Light Forcing achieves up to **4.71×** attention speedup and **2.54×** DiT speedup. We will include a more detailed profiling analysis in the next version.
>
>
>     |Type|Resolution|5s|10s|15s|
>     |---|---|---|---|---|
>     |Peak memory (GB)|480p|23.18|28.99|34.70|
>     |KV memory (GB)|480p|6.04|12.08|18.11|
>     |Peak memory (GB)|720p|36.10|49.52|62.96|
>     |KV memory (GB)|720p|13.93|27.87|41.80|
>
>     |Method|Resolution|Duration (s)|Attention latency (ms)|Speedup|DIT latency (ms)|Speedup|
>     |---|---|---|---|---|---|---|
>     |Dense|480p|5|777|1.00×|1244|1.00×|
>     |Light Forcing|480p|5|217|3.58×|685|1.82×|
>     |Dense|480p|10|1515|1.00×|1982|1.00×|
>     |Light Forcing|480p|10|407|3.73×|874|2.27×|
>     |Dense|480p|15|2239|1.00×|2708|1.00×|
>     |Light Forcing|480p|15|591|3.79×|1060|2.55×|
>     |Dense|720p|5|2752|1.00×|3570|1.00×|
>     |Light Forcing|720p|5|584|4.71×|1402|2.54×|
> - **Preliminary results on larger models**. Thank you for the suggestion. We conduct a preliminary study on Krea Realtime 14B [4]. Due to GPU resource constraints, we implement the two key modules in our paper, CAG and HSA, in a **training-free manner**. As shown in the table below, our method still substantially outperforms naive sparse attention, which suggests that Light Forcing generalizes robustly to larger models.
>
>
>     |Method|Quality Score ↑|Semantic Score ↑|Total Score↑|
>     |---|---|---|---|
>     |Dense|84.6|81.4|84|
>     |Naive Dynamic Sparse Attention|82.5|81.7|82.4|
>     |+CAG|83.4|81|83|
>     |+CAG+HSA|83.6|81.1|83.2|
>
> [4] https://github.com/krea-ai/realtime-video

---

> > ### Author Rebuttal · Reviewer_3qfq · 2026-04-03
> >
> > Thank you for the authors’ rebuttal. I will update my score accordingly.

---

> > > ### Author Response · Authors · 2026-04-03
> > >
> > > Dear Reviewer 3qfq,
> > >
> > > We appreciate your encouraging feedback that our responses addressed your concerns and that you decided to update the score. We noticed that the score has **not** been raised accordingly, and we were wondering if you would consider updating it to reflect your current evaluation.
> > >
> > > If there are still any concerns that have not been fully addressed, we would be very happy to continue the discussion. We would also greatly appreciate it if you would consider raising your score.
> > >
> > > Best regards,
> > >
> > > Authors

---

### Official Review · Reviewer_VPPV · 2026-03-11

**Soundness:** 3
**Presentation:** 3
**Significance:** 3
**Originality:** 3
**Overall Recommendation:** 4
**Confidence:** 3

**Summary:**

This paper introduces Light Forcing, a sparse attention mechanism tailored for autoregressive (AR) video generation. The authors propose two main components to mitigate the quadratic complexity of attention: a Chunk-Aware Growth mechanism that progressively allocates higher sparsity to later chunks, and a coarse-to-fine dynamic mask selection strategy designed to capture long-range context. The framework is evaluated on 1.3B-parameter AR video generation models.

**Compliance With Llm Reviewing Policy:**

Affirmed.

**Final Justification:**

My concerns have been adequately addressed.

**Key Questions For Authors:**

While I fully understand that access to diverse or enterprise-grade hardware may be limited, evaluating FA2 on the 5090 GPU provides a somewhat questionable baseline. FA2 is not heavily optimized for this specific GPU architecture, meaning the dense baseline might be artificially slow and thus inflate the reported relative speedups of the sparse methods. Although FA3 is also not natively ready for the 5090, I am curious how the latency improvements of the proposed method would hold up when compared against a truly architecture-aligned dense baseline (e.g., FA3 on Hopper, or FA2 on older architectures such as Ampere/Ada).

**Limitations:**

No.

See the weaknesses and questions.

**Strengths And Weaknesses:**

Strengths
* The paper is well-motivated. The core insight that early generation stages require dense attention while later stages can be sparse is reasonable and intuitive, and the proposed method is logical and easy to follow.
* The experimental section is comprehensive.

Weaknesses
* The proposed method relies on the off-the-shelf SparseAttn kernel, and its contribution is largely limited to a block selection policy, which lacks significance.
* The method requires post-training, so it is unfair to compare it with training-free approaches (e.g., SVG2).

---

> ### Author Rebuttal · Authors · 2026-03-31
>
> Thanks to the reviewer for the constructive comments.
>
> - **Significance beyond kernel dependency**. Thank you for the comment. We would like to clarify that Light Forcing mainly focuses on algorithmic innovation in sparse attention (*CAG allocates higher attention budgets to earlier chunks and progressively decay for later chunks while HSA captures global and local dependencies via coarse-to-fine frame and block selection*), our method is **not limited** to SpargeAttn [1] kernel. We use SpargeAttn only for practical latency evaluation, while Light Forcing is compatible with *any* block-wise sparse attention kernel, *e.g.*, FlashInfer [2], MagiAttention [3], and SpargeAttn[1]. Moreover, we also make several kernel-level improvements to support training and deployment: 1) a Triton backward implementation for sparse attention training, 2) a sparse kernel that supports different shapes for q and kv, and 3) a more efficient Triton implementation of block selection in HSA. We will also provide a more detailed description of these kernel-level optimizations in the next version.
>
> [1] Spargeattn: Accurate sparse attention accelerating any model inference. ICML 2025.
>
> [2] FlashInfer: Efficient and Customizable Attention Engine for LLM Inference Serving.
>
> [3] MagiAttention: A Distributed Attention Towards Linear Scalability for Ultra-Long Context, Heterogeneous Mask Trainin.
>
> - **Unfair comparison with training-free methods**. We would like to clarify that our comparison is **not** limited to training-free baselines such as SVG2. We also compare with **two state-of-the-art finetunable** methods, *i.e.*, **VMoBA [4] and SLA [5]**. For a fair comparison, all three methods, together with Light Forcing, are post-trained for 2k iterations. More details can be found in Section 5.1, Experimental Details (lines 303–307) of the paper. As shown in Table 1 in paper, Light Forcing outperforms these finetunable methods on both model families, which further demonstrates its superiority.
>
> [4] Vmoba: Mixture-of-block attention for video diffusion models.
>
> [5] Sla: Beyond sparsity in diffusion transformers via fine-tunable sparse-linear attention. ICLR 2026.
>
> - **Hardware-specific baseline concern**.  Thank you for the thoughtful comment. We fully understand the concern that the dense baseline on RTX 5090 may not be perfectly architecture-aligned. There are three main reasons for our current evaluation setup.
>     - Our work mainly focuses on the algorithmic acceleration of sparse attention for autoregressive video generation, and can be readily combined with different sparse attention kernels. When Light Forcing is paired with a well-optimized kernel, the practical speedup can be much closer to the theoretical one.
>     - Our latency evaluation relies on SparseAttention, which is built on top of *FlashAttention 2 and is not heavily optimized for Hopper architectures*. Therefore, a direct comparison with FlashAttention 3 may itself be unfair. We also acknowledge this point in the limitation section of the paper (**lines 709–713**). Due to time constraints, we have not yet completed sparse attention kernel optimization based on FlashAttention 3, but we will include such comparisons in a future version.
>     - To further address this concern, we also conduct detailed latency evaluations on a single **A100 GPU**. For 5-second 480p video generation, Light Forcing achieves **1.22×** end-to-end speedup and **2.56×** attention speedup. As the token sequence length increases, *i.e.*, with *longer generation duration or higher resolution*, the benefit becomes more pronounced. Notably, for 720p video generation, Light Forcing reaches **1.51×** end-to-end speedup and **3.82×** attention speedup. This further demonstrates that Light Forcing consistently provides acceleration across different GPU platforms, highlighting its broad applicability. Thank you for the suggestion! We will include more results on different GPU architectures in the next version.
>
>     | Type | Resolution | Duration (s) | e2e Latency (s) | Speedup | Attention Latency (s) | Speedup |
>     | --- | --- | --- | --- | --- | --- | --- |
>     | FA2 | 480p | 5 | 11.48 | 1.00× | 3.38 | 1.00× |
>     | Light Forcing | 480p | 5 | 9.42 | 1.22× | 1.32 | 2.56× |
>     | FA2 | 480p | 15 | 39.66 | 1.00× | 14.95 | 1.00× |
>     | Light Forcing | 480p | 15 | 30.11 | 1.32× | 5.40 | 2.77× |
>     | FA2 | 720p | 5 | 31.25 | 1.00× | 14.26 | 1.00× |
>     | Light Forcing | 720p | 5 | 20.73 | 1.51× | 3.74 | 3.82× |

---

> > ### Author Rebuttal · Reviewer_VPPV · 2026-04-01
> >
> > Thanks for the authors` rebuttal, I will raise my score.

---

### Official Review · Reviewer_Yi7m · 2026-03-13

**Soundness:** 3
**Presentation:** 3
**Significance:** 3
**Originality:** 2
**Overall Recommendation:** 5
**Confidence:** 5

**Summary:**

This paper studies how to speed up autoregressive video diffusion with sparse attention, arguing that sparse methods designed for bidirectional video generation degrade in the autoregressive setting because they ignore the unequal importance of different chunks and discard useful long-range context. The authors propose two parts: Chunk-Aware Growth, which gives earlier chunks more attention budget and later chunks more sparsity, and Hierarchical Sparse Attention, which retrieves relevant past frames and then relevant blocks within those frames in a coarse-to-fine way.

**Compliance With Llm Reviewing Policy:**

Affirmed.

**Final Justification:**

Thanks the reviewers for their rebuttal efforts. I do not have much issue with the paper initially and will maintain my score.

**Key Questions For Authors:**

See weaknesses.

**Limitations:**

yes

**Strengths And Weaknesses:**

Strengths:
- The paper targets a practically important bottleneck: attention becomes a dominant share of inference cost in autoregressive video generation, and the problem is well motivated from a deployment perspective.
- The method does not look overly brittle in the limited sensitivity study: changing HSA top-k from 6 to 9 or 12 only slightly changes the reported scores.
- The ablation is useful and supports the paper’s story.

Weaknesses:
- Despite the long-horizon motivation, the main benchmark setup is still 5-second video generation. That makes the evidence for truly long-horizon consistency weaker than the framing suggests -- on 5-sec videos, we usually don't see much content especially dynamics anyways.
- The comparison set is somewhat imperfect because the paper itself notes that most baselines are sparse attention methods originally designed for bidirectional video generation, then adapted to autoregressive models. That is still useful though.
- The strongest deployment headline is not solely due to the proposed sparse attention method; it also relies on LightX2V, LightVAE, and FP8 quantization. The paper could do a better job separating what comes from Light Forcing itself versus the extra engineering stack.
- The paper has great related work section but could benefit from a short discussion with the following methods too:
  - One-Minute Video Generation with Test-Time Training
  - Mixture of Contexts for Long Video Generation
  - Pack and Force Your Memory: Long-form and Consistent Video Generation

---

> ### Author Rebuttal · Authors · 2026-03-31
>
> Thanks to the reviewer for the constructive comments.
>
> - **Long-horizon consistency**. Thank you for the comment. We would like to clarify that, since Self Forcing does **not** natively support generation beyond 5 seconds, we evaluate Light Forcing with Infinite-Forcing [1] on 15-second videos using VBench-Long [2]. As shown in the table below, Light Forcing still maintains competitive total score (*83.6 vs. 82.5*), suggesting that Light Forcing remains stable for longer generation. Beyond the overall score, we also report Subject Consistency, Background Consistency, Overall Consistency, Motion Smoothness, and Dynamic Degree. In particular, Light Forcing achieves a higher Dynamic Degree than dense attention (*61.2 vs. 38.6*), which further indicates that it can effectively support long-video generation in *single-scene settings*. We view extending this advantage to more complex *multi-scene long-video generation* as an important direction for future work.
>
>
>     | Infinite Forcing | Quality Score ↑ | Semantic Score ↑ | Total Score↑ |
>     | --- | --- | --- | --- |
>     | Dense Attention | 83.2 | 79.4 | 82.5 |
>     | Light Forcing | 84.6 | 79.6 | 83.6 |
>
>     | Infinite Forcing | Subject Consistency | Background Consistency | Overall Consistency | Motion Smoothness | Dynamic Degree |
>     | --- | --- | --- | --- | --- | --- |
>     | Dense Attention | 98.5 | 97.7 | 26.1 | 98.9 | 38.6 |
>     | Light Forcing | 98.2 | 97.3 | 26.2 | 98.8 | 61.2 |
>
> [1] Infinite-Forcing: Towards Infinite-Long Video Generation.
>
> [2] Vbench++: Comprehensive and versatile benchmark suite for video generative models. *TPAMI* 2025.
>
> - **Imperfect baseline comparison**. Thank you for pointing this out. We would like to clarify that, to the best of our knowledge, there is currently no prior sparse attention method specifically designed for autoregressive video generation, and our work is a pioneer work in this direction. Existing sparse attention methods for bidirectional video generation are still meaningful baselines, since they are generally effective, especially under *relatively low sparsity ratios*. In contrast, Light Forcing is specifically designed to account for the characteristics of autoregressive video generation (*Sections 4.1 and 4.2*), which enables it to maintain better accuracy under much higher sparsity ratios (**around 90%**). We also include additional efficient video generation baselines, including cache-based methods (*e.g.*, TeaCache) and token reduction methods (*e.g.*, DyCoke), which further support the superiority of Light Forcing (see our response to reviewer **3qfq** on **Incomplete baselines among AR acceleration methods**).
> - **Separation of method contribution and system optimizations**. Thank you for the comment. We would like to clarify that the use of LightVAE and FP8 quantization is not our contribution, but is included to show that Light Forcing is compatible with orthogonal efficient methods and can be combined with them for greater practical speedup. Our core goal is still to accelerate attention. In our experiments, Light Forcing achieves up to **3.29×** attention speedup (Figure 6), even under relatively low-resolution settings. As the token sequence length increases, *e.g.*, with *higher resolutions, longer generation durations, or with further optimized kernel implementations*, the acceleration benefit is expected to become more pronounced. For example, as shown in the table below, for the 1.3B Self Forcing model at 720p, Light Forcing already delivers **1.53×** end-to-end speedup and **4.03×** attention speedup on RTX 5090 for 5-second video generation. Thank you for the comment. We will also provide a clearer explanation of this part in the future version, to better distinguish Light Forcing itself from the extra engineering stack.
>
>
>     | Type | e2e Latency (s) | Speedup | Attention Latency (s) | Speedup |
>     | --- | --- | --- | --- | --- |
>     | Dense Attention | 24.16 | 1.00× | 11.13 | 1.00× |
>     | Light Forcing | 15.79 | 1.53× | 2.76 | 4.03× |
> - **Additional related work discussion**. Thank you for the positive feedback on our related work section and for suggesting these valuable references. These three papers are indeed highly relevant: (1) *One-Minute Video Generation with Test-Time Training* introduces test-time training layers into a pre-trained model and uses expressive hidden states to model extremely long video contexts. (2) *Mixture of Contexts for Long Video Generation* proposes a learnable sparse routing mechanism that makes long-range memory more scalable and consistent. (3) *Pack and Force Your Memory: Long-form and Consistent Video Generation* presents MemoryPack, a retrieval-based memory design that combines text and image guidance for long-form video generation. We appreciate these recommendations and will include a discussion of them in the related work section of the next revision.

---

> > ### Author Rebuttal · Reviewer_Yi7m · 2026-04-01
> >
> > Thanks the reviewers for their rebuttal efforts. I do not have much issue with the paper initially and will maintain my score.

---

### Official Review · Reviewer_tAhP · 2026-03-13

**Soundness:** 3
**Presentation:** 3
**Significance:** 2
**Originality:** 3
**Overall Recommendation:** 4
**Confidence:** 3

**Summary:**

This paper proposes light forcing, a sparse attention framework designed for autoregressive (AR) video diffusion models. It introduces two mechanisms: CAG allocates lower sparsity to earlier chunks based on a TV-bound-inspired error analysis, and HSA retrieves informative historical context via coarse-to-fine frame-then-block selection. Experiments on Self Forcing and LongLive (both 1.3B) show that light forcing matches or surpasses dense attention on VBench (84.5 vs 84.1) with 1.3x end-to-end speedup, and reaches 19.7 FPS on RTX 5090 when combined with FP8 and LightVAE.

**Compliance With Llm Reviewing Policy:**

Affirmed.

**Key Questions For Authors:**

- How sensitive is the method to the block size (the value is fixed at 64 in the paper)? An ablation over block sizes (e.g.32, 128) would clarify the generality of the approach.
- Have alternative sparsity scheduling functions (eg. linear or logarithmic decay) been compared against the current form?
- For early chunks where the number of past frames is smaller than or comparable to top-k, is the frame-selection stage in HSA effectively bypassed?

**Limitations:**

Yes

**Strengths And Weaknesses:**

Strengths:
- The chunk-aware sparsity allocation is a novel and well-motivated idea. The toy experiment clearly shows that early-chunk sparsity causes irreversible quality loss while late-chunk sparsity is nearly lossless, providing strong intuition for the design.
- The hierarchical frame-then-block selection in HSA is well-justified by the attention logit analysis and shows the critical historical context varies across layers, heads, and timesteps in ways that sliding windows cannot capture.
- The ablation study cleanly isolates the contribution of each component, showing that CAG improves static quality and HSA restores dynamics, with the full system exceeding dense attention.
- The validation on two types of independent AR models demonstrates generalizability of light forcing.

Weaknesses:
- The standalone speedup from sparse attention is modest (1.19x-1.30x). The headline 19.7 FPS and 2.33x speedup rely on orthogonal techniques (FP8, LightVAE) that are not contributions of this paper.
- The theoretical grounding involves heuristic simplifications. The mapping from TV bounds to the sparsity formula treats logarithmic factors as constants and the functional form is not formally justified for the sparse attention setting.
- All experiments use 1.3B models and 5-second videos only. Whether CAG's progressive sparsity remains stable for longer generation or scales to 14B (and larger) models is unknown.

---

> ### Author Rebuttal · Authors · 2026-03-31
>
> Thanks to the reviewer for the constructive comments.
>
> - **Modest standalone speedup.** First, we would like to clarify that the core goal of our paper is to achieve extreme attention acceleration. In our experiments, Light Forcing achieves up to **3.29×** attention speedup (Figure 6 in paper), even under relatively low-resolution settings. As the token sequence length increases, *e.g.*, with *higher resolutions, longer generation durations, or with further optimized kernel implementations*, the acceleration benefit is expected to become more pronounced. As shown in the table below, for the 1.3B Self Forcing model at 720p, Light Forcing already delivers **1.53×** end-to-end speedup and **4.03×** attention speedup on RTX 5090 for 5-second video generation. Therefore, the improvement on the main bottleneck (*i.e.*, attention) is already substantial, and becomes even more pronounced as the token sequence length increases. Finally, Figure 6 also shows that our method is compatible with other orthogonal acceleration techniques, such as near-lossless FP8 quantization and efficient VAE, which together lead to more significant overall speedups in practice. This composability is an additional advantage of our method.
>
>
>     |Type|e2e Latency (s)|Speedup|Attention Latency (s)|Speedup|
>     |---|---|---|---|---|
>     |Dense Attention|24.16|1.00×|11.13|1.00×|
>     |Light Forcing|15.79|1.53×|2.76|4.03×|
> - **Scalability to longer videos and larger models**. We further validate CAG on both longer videos and a 14B model. For longer videos, since *Self Forcing does not natively support generation beyond 5 seconds*, we evaluate Light Forcing with Infinite-Forcing [1] on 15-second videos using VBench-Long [2]. As shown in the table below, Light Forcing still maintains competitive Total Score (*83.4 vs. 82.5*), suggesting that the progressive sparsity remains stable for longer generation. For larger models, we additionally evaluate Krea Realtime 14B [3] on 5-second videos. Due to GPU resource constraints, we conduct a **training-free** comparison among dense attention, naive dynamic sparse attention, and CAG. While CAG is slightly below dense attention, it still clearly outperforms naive dynamic sparse attention (*83.0 vs. 82.4*), indicating that CAG remains effective at the 14B scale even without additional training.
>
>
>     |Infinite Forcing|Quality Score ↑|Semantic Score ↑|Total Score↑|
>     |---|---|---|---|
>     |Dense Attention|83.2|79.4|82.5|
>     |+CAG|84.5|79|83.4|
>
>     |Krea Realtime 14B|Quality Score ↑|Semantic Score ↑|Total Score↑|
>     |---|---|---|---|
>     |Dense Attention|84.6|81.4|84.0|
>     |Naive Dynamic Sparse Attention|82.5|81.7|82.4|
>     |+CAG|83.4|81.0|83.0|
>
> [1] Infinite-Forcing: Towards Infinite-Long Video Generation.
>
> [2] Vbench++: Comprehensive and versatile benchmark suite for video generative models. *TPAMI* 2025.
>
> [3] https://github.com/krea-ai/realtime-video
>
> - **Sensitivity to block size**. We evaluate the trained Light Forcing model (*trained with block size = 64*) using different block sizes at inference. As shown in the table below, different block sizes all yield competitive accuracy (*84.4–84.6*), while the main difference is in the Dynamic Degree (*65.6–69.4*). These results suggest that Light Forcing is not sensitive to the block size and generalizes well across different block configurations.
>
>
>     |Blocksize|Quality Score ↑|Semantic Score ↑|Total Score↑|Dynamic Degree|
>     |---|---|---|---|---|
>     |32|85.5|81|84.6|69.4|
>     |64 (reported in paper)|85.4|80.9|84.5|66.7|
>     |128|85.3|81|84.4|65.6|
> - **Heuristic theoretical simplification & Alternative sparsity scheduling**. In Eq. 4, the dominant term is $\frac{1}{\sqrt{t}}$, while the logarithmic factors are higher-order terms and have limited numerical impact in practice. Our current formulation is therefore intended to capture the main trend. We appreciate this important point and will further investigate a more formal theoretical exploration in the next version. We further compare CAG with a linear decay schedule. For a fair comparison, we keep the total attention FLOPs across chunks the same for different schedules (following Eq. 6 in the paper). Concretely, we set the sparsity ratio of the first chunk to 0.82 and increase it by 0.02 for each subsequent chunk. As shown in the table below, the linear decay schedule performs better than a fixed sparsity schedule, but still underperforms CAG.
>
>
>     |Method|Quality Score ↑|Semantic Score ↑|Total Score↑|
>     |---|---|---|---|
>     |fixed|83|82.1|82.8|
>     |linear decay|83.2|82|83|
>     |CAG|83.4|82.3|83.2|
> - **Early-chunk behavior of HSA**. Yes. When the number of past frames is small, we consider all of them important and allow them to participate in the attention computation. Therefore, the frame-selection stage in HSA is only applied when the number of past frames exceeds top-k.

---

> > ### Author Rebuttal · Reviewer_tAhP · 2026-04-02
> >
> > My concerns have been resolved, and I will maintain the rating.

---

### Decision · Program_Chairs · 2026-04-30

**Decision:**

Accept (regular)

**Comment:**

The overall review signal is positive. While some limitations remain, particularly in the breadth of the evaluation and the still limited evidence at larger scales, the paper makes a meaningful and well-executed contribution to sparse attention in autoregressive video generation. I therefore recommend accept.